# In vivo tumor immune microenvironment phenotypes correlate with inflammation and vasculature to predict immunotherapy response

Response to immunotherapies can be variable and unpredictable. Pathology-based phenotyping of tumors into 'hot' and 'cold' is static, relying solely on T-cell infiltration in single-time single-site biopsies, resulting in suboptimal treatment response prediction. Dynamic vascular events (tumor angiogenesis, leukocyte trafficking) within tumor immune microenvironment (TiME) also influence anti-tumor immunity and treatment response. Here, we report dynamic cellular-level TiME phenotyping in vivo that combines inflammation profiles with vascular features through non-invasive reflectance confocal microscopic imaging. In skin cancer patients, we demonstrate three main TiME phenotypes that correlate with gene and protein expression, and response to toll-like receptor agonist immune-therapy. Notably, phenotypes with high inflammation associate with immunostimulatory signatures and those with high vasculature with angiogenic and endothelial anergy signatures. Moreover, phenotypes with high inflammation and low vasculature demonstrate the best treatment response. This non-invasive in vivo phenotyping approach integrating dynamic vasculature with inflammation serves as a reliable predictor of response to topical immune-therapy in patients.

Immunotherapy, especially immune checkpoint blockade therapy, has revolutionized cancer management by providing durable responses in several cancers[1–3]. However, only a subset of patients derives long-term benefit, highlighting a clinical need to develop effective biomarkers for patient stratification[4–6]. Phenotyping of tumors into 'hot', 'cold' or 'altered' based on infiltration of CD3+ and CD8+ T-cells at tumor center and margin (Immunoscore)[7], PD-1/PD-L1 expression, and tumor mutation burden are important determinants of response to immunotherapy in solid cancers[8,9]. Although hot versus cold tumor phenotypes have shown association with treatment response and overall cancer outcomes, specific immune cell subsets modify this association, including regulatory T cells (Tregs), myeloid-derived suppressor cells (MDSCs), and tumor-associated macrophages (TAMs)[10]. Importantly, not all hot phenotypes respond to treatment, suggesting that immune cell infiltration is important, but not always sufficient, for inducing potent anti-tumor immunity to eradicate tumors[11,12]. Evidently, tumors utilize additional mechanisms for evading immune response while establishing an immune-suppressive microenvironment, complicating patient stratification strategies because of dynamic tumor host/immune crosstalk and tumor-intrinsic biology[8,13,14].

The tumor vasculature is a key component of the microenvironment that can influence tumor behavior and treatment response. Tumor vasculature plays a central role in T-cell trafficking via regulation of endothelial adhesion molecule expression and creation of immunosuppressive microenvironments[15]. Angiogenesis promotes immune evasion through induction of a highly immunosuppressive TiME by inhibiting dendritic cell (DC) maturation, T-cell development

✉ e-mail: aditisahu@gmail.com; rajadhym@mskcc.org

and function, and most importantly, limiting access of effector immune cells to tumors by modulating leukocyte trafficking[16]. In addition, tumor vasculature can display decreased expression of adhesion molecules, and demonstrate non-responsiveness to inflammatory cytokines through development of vascular endothelial anergy. By downregulating trafficking of effector cells and upregulating trafficking of regulatory immune cells, endothelial anergy contributes to ineffective anti-tumor immune responses and immune evasion[17-19].

To address the complex and highly interdependent vascular-inflammation axis within the TiME, in vivo phenotyping that integrates dynamic vascular features with inflammation may be more advantageous as compared to ex vivo pathological phenotyping based mainly on infiltrating T-cells. High-resolution non-invasive in vivo imaging is fundamental to this integrative phenotyping, since static ex vivo analyses on patient tissue are limited in recapitulating dynamic vascular events and the continuous, evolving cellular-level crosstalk between the immune system and tumor[6,20]. Reflectance confocal microscopy (RCM) is a high-speed (pixel times -0.10 μs, frame rates 10–30 per second) cellular-level label-free imaging approach based on backscattered light and endogenous tissue contrast, capable of capturing dynamic phenomena inside patients in real-time[21,22]. Further, large image mosaics (64 mm² in 50 s) imaged to a depth of -0.25 mm can facilitate spatial resolution of features. RCM is routinely used for real-time skin cancer diagnosis and management at the bedside[23,24]. Although a few studies have reported RCM imaging of vessels and leukocyte trafficking in humans[25-27], analysis of individual RCM TiME

features, the vascular-inflammation axis and its role in treatment response prediction have not been studied.

In this work, we report TiME phenotypes detected in vivo in basal cell carcinoma (BCC) and melanoma. We define TiME phenotypes with key inflammation and vasculature features such as vessel diameter, vessel density, leukocyte trafficking, intratumoral inflammation, peritumoral inflammation and perivascular inflammation. We demonstrate feasibility of automated quantification of immune and vascular features. Further, we also perform in-depth histopathological validation and molecular correlation of TiME features and phenotypes with gene and protein expression. Finally, we investigate the utility of immune and vascular features and TiME phenotypes in predicting response to a toll-like receptor agonist (TLRA) topical immune-therapy in a prospective pilot study (Fig. 1).

## Results

### Immune and vascular features on RCM correlate with histopathology and demonstrate unique TiME phenotypes

TiME phenotyping was investigated by integrating inflammation and vasculature features. First, agreement between RCM and histopathological TiME features was explored. Manual evaluation of RCM TiME features (Fig. S1a–c, Supplementary Movies 1–3) by two independent readers resulted in substantial to almost perfect agreement (k = 0.62–1.0) for most RCM features. No TiME features were observed in the normal perilesional area (Fig. S1d); thus analysis was restricted to tumor lesion. RCM manual evaluation correlated well with corresponding histopathological features evaluated by a board-certified

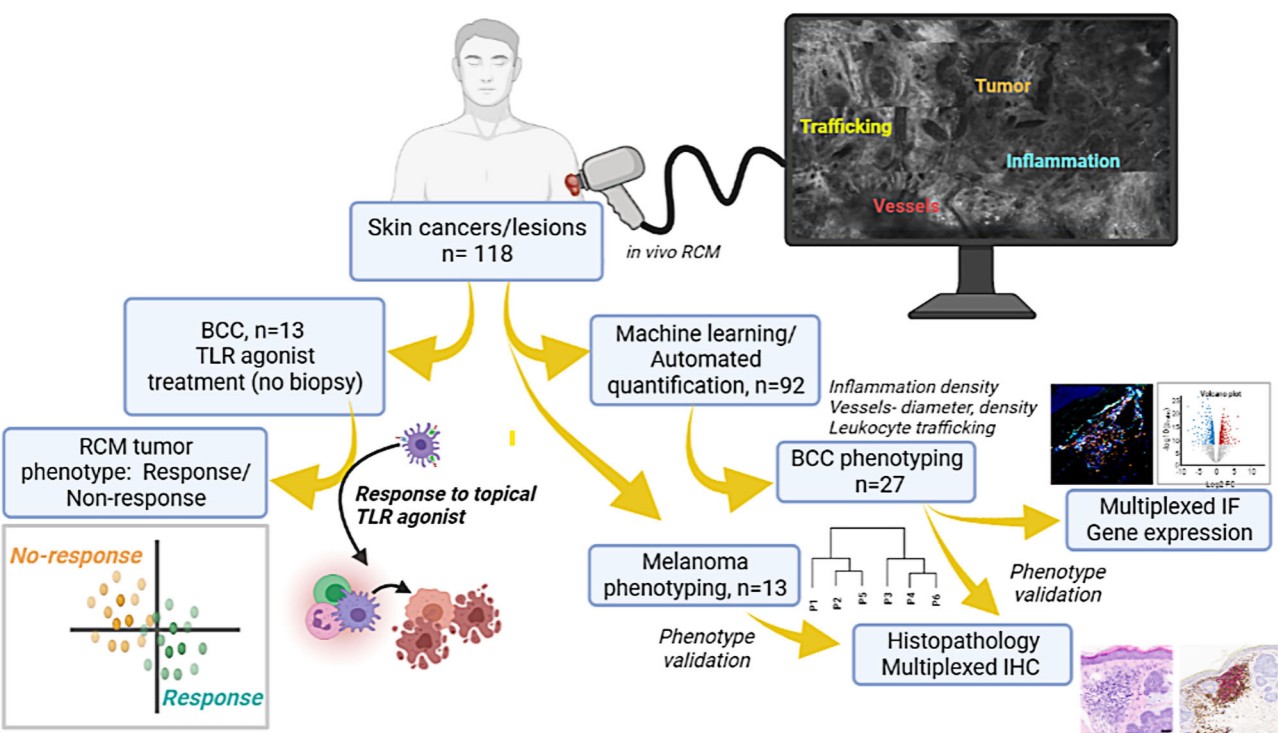

**Fig. 1 | TiME phenotypes were derived from RCM imaging and correlated with underlying biology and treatment response to topical toll-like receptor agonist (TLRA) imiquimod.** In vivo RCM imaging was performed on patients with clinically suspected skin cancers or rashes visiting the Dermatology Service at MSKCC. Imaging on the lesion was performed to span large field-of-view (FOV) for exhaustive sampling in tumor, peritumoral and adjacent normal areas. Tumor, inflammation, vasculature and trafficking were imaged within each lesion. Data was used for machine learning and automated quantification of inflammation density, vessel diameter and density, and frequency of leukocyte trafficking. RCM-TiME phenotypes were investigated using unsupervised analysis for basal cell carcinoma (BCC), and melanoma cohorts. The RCM phenotypes were correlated with

pathology and dual immunohistochemistry (IHC) for CD3+ (T-cell) and CD20+ (B-cell) labeling of tertiary lymphoid structures. BCC phenotyping was additionally validated using multiplexed immunofluorescence (CD8+, FOXP3, CD68+, PD-1+, PD-L1+) and bulk RNA sequencing. A subset of patients with confirmed BCC diagnoses on RCM undergoing treatment with a TLRA agonist were enrolled. The patients were imaged 6 months after end of treatment to confirm tumor clearance and were classified as responders (complete tumor regression) or non-responders (incomplete or no tumor regression). Treatment response was correlated with TiME features and phenotypes. Created with www.Biorender.com RCM: reflectance confocal microscopy; BCC: basal cell carcinoma; IHC: immunohistochemistry; TLRA: toll-like receptor agonist.

**Table 1 | High agreement suggests the reproducibility and presence of TiME features on histopathology**

| TiME feature | RCM 2-reader agreement (Cohen's kappa) | Histology vs. average 2- reader (Binary) |
|---|---|---|
| Number of vessels | 0.62 | 0.84 |
| Dilated vessel presence | 0.72 | 0.89 |
| Trafficking | 0.76 | 0.72 |
| Intratumor inflammation | 1.00 | 1.00 |
| Peritumor inflammation | NA | 0.97 |
| Mucin | 0.93 | 0.74 |
| Perivascular inflammation | NA | 1.00 |

Substantial to almost perfect agreement (k = 0.62-1.0) was observed for RCM features in n = 27 BCC lesions between 2-readers, k for peritumor inflammation and perivascular inflammation agreement could not be computed due to 100% prevalence. Good to very good agreement (AC$_1$: 0.74–1.0) was found in the binary analysis between average RCM evaluation and dermatopathologist grading of features.

dermatopathologist (Table 1), demonstrating good to very good agreement (AC$_1$: 0.74–1.0). Unsupervised multivariate clustering of TiME features in BCC patients revealed three main phenotypes with variable prevalence of inflammation (Inflam) and vascular (Vasc) features (Fig. 2a). This feature heterogeneity was clearly apparent in the varying degrees of inflammation, trafficking and vessels in corresponding RCM and H&E images of these phenotypes. No correlation of TiME phenotypes with clinical factors such as BCC subtype, gender, age, sun exposure and ulceration were observed (Fig. 2a). Phenotypic classes were assigned based on individual TiME measurement contributions to the first two principal components, which together comprise 77% of the total variability (Fig. 2b). Number of vessels, trafficking and dilated vessels were major contributors to the variability captured by the PC1 or dimension 1 (Fig. 2c) whereas perivascular, peritumor and intratumor inflammation were major contributors to variation captured by PC2 or dimension 2 (Fig. 2d). TiME feature contributions to PC1 and PC2 therefore explain the distribution of BCC patient samples according to variability in inflammation and vasculature, where samples along PC1 cluster based on vasculature (Fig. 2e, f, low=purple, high=pink and black) and samples distributed along PC2 cluster based on inflammation (Fig. 2e, f low=black, high=pink and purple) thereby allowing for discrete phenotypes to emerge: Inflam$^{LOW}$Vasc$^{HIGH}$ (black), Inflam$^{HIGH}$Vasc$^{LOW}$ (purple) and Inflam$^{HIGH}$Vasc$^{HIGH}$ (pink).

**Molecular landscape associates with Inflam$^{HIGH}$ and Vasc$^{HIGH}$ signatures**

To determine transcriptional variation underlying observed phenotypic differences in inflammation and vasculature, we measured transcript abundance by bulk RNA sequencing from 14 BCC samples representing the two groups Inflam$^{HIGH}$Vasc$^{HIGH}$ and Inflam$^{LOW}$Vasc$^{HIGH}$ (Fig. S2a–d). To identify genes specific to functional gene regulatory networks relevant to BCC tumor and TiME, we performed modular co-expression analyses using CEMiTool (Fig. S2e–h)[28]. Of the resulting 8 modules comprised of co-regulated genes, 2 modules (M2 and M5) showed statistically significant enrichment of gene activity in RCM TiME phenotypes, both M2 and M5 displayed higher activity in the Inflam$^{HIGH}$ group (Fig. S3a–d and Fig. 3a–b). To understand differential control of biological processes and cell types, M2 and M5 were annotated using gene ontology (GO) enrichment analysis. We discovered distinct biological processes, pathways, and cell types for M2 compared to M5; M2 module genes exhibited significant enrichment in pathways such as pro-inflammatory signatures, allograft rejection, interferons, and myeloid cells while genes comprised in M5 displayed

signatures related to leukocyte adhesion and migration along with cytokine receptor activity. M2 shows signatures enriched in cells comprising blood whereas genes expressed in M5 were associated with blood vessels (Fig. 3c and Fig. S3e). Given the shared concentration of genes participating in immune cell function, we generated a gene interaction network using gene connections defined in T-lymphocytes (curated gene pairs downloaded from TissueNexus[29], see methods). We discovered module hub genes participate in T-lymphocyte gene regulatory networks (GRNs) to a high degree and together with network hub genes are connected to genes enriched in GO terms involved in regulation of distinct inflammatory pathways for M2 and M5. For M2 T-lymphocyte network, these pathways include T-cell activation and differentiation, myeloid cell differentiation and leukocyte adhesion. In contrast, M5 T-lymphocyte networks were enriched in blood vessel proliferation and cell adhesion (Fig. 3d and Fig. S3f). We next assessed gene networks in tissues and cell types related to BCC and TiME albeit beyond T-lymphocytes. Interestingly, we find module hub genes persist in connectivity with gene networks in skin, macrophage and blood. Notably, network hub genes are largely shared across cell types, indicating M2 and M5 genes participate in shared pathways across functionally distinct tissues and contribute to observed differences in inflammatory phenotypes. Indeed, module hub genes (*SLA, DOCK2, CD34, ABCA6/9*) and shared network hub genes including *ICAM1, VCAM1, TGFBR3, CXCL12* and *PDGFD* are known to be involved in immune and vascular signaling and function, immune cell migration and all of these are overexpressed in the Inflam$^{HIGH}$ as compared to the Inflam$^{LOW}$ phenotype (Fig. S3g and Fig. 3e).

Since a role for differential regulation of immune cell function among RCM phenotypes emerged from assessment of bulk RNA-seq, we hypothesized variations in populations of immune cells could contribute to differences in RCM phenotypes. To assign transcripts to immune cell types and estimate cell proportions, we deconvoluted bulk RNA-seq using CIBERSORTx[30] (Fig. 3f). Unsupervised k-means clustering on CIBERSORTx output distinguished Inflam$^{HIGH}$ and Inflam$^{LOW}$ phenotypes, which was not achieved using all variable transcripts from bulk RNA-seq or when assessing variability in genes comprising M2 and M5 (Fig. 3g, Figs. S2i–k, S3h). Relative differences in composition of major immune cell types were inferred and significant differences in cell proportion across phenotype groups were discovered for CD4$^+$ T memory cells (resting) and M1 macrophages (Fig. 3h).

**Inflam$^{HIGH}$ phenotypes correlate with abundant CD8$^+$ and CD8$^+$ PD1$^+$ cells**

The gene expression results identified immune cell composition and function, especially T-cells and macrophages were important in distinguishing RCM phenotypes. Subsequently, differences in T-cells and macrophages along with PD-1/PDL-1 checkpoint expression were investigated across phenotypes in BCC patients. Assessing peritumoral areas of inflammation, median % (mean %, 95% CI) proportions of CD8$^+$ cells were 6.35% (12.01%, 1.53–22.49%) in Inflam$^{LOW}$Vasc$^{HIGH}$, 21.97% (21.24%, 15.68–26.79%) in Inflam$^{HIGH}$Vasc$^{LOW}$ and 11.74% (11.2%, 8.1–14.29%) in Inflam$^{HIGH}$Vasc$^{HIGH}$ groups. Furthermore, out of these CD8$^+$ cells, 25.22% (27.33%, 13.11–41.55%), 46.86% (40.7%, 23.99–57.41%), 31.01% (31.04%, 20.01–42.06%) were PD1 positive in Inflam$^{LOW}$Vasc$^{HIGH}$, Inflam$^{HIGH}$Vasc$^{LOW}$ and Inflam$^{HIGH}$Vasc$^{HIGH}$, respectively (Fig. 4a). Similar trends were seen in FOXP3$^+$ T-regulatory cells and CD68$^+$ macrophages in peritumoral infiltrates (Fig S4a). High intratumoral infiltration by CD68$^+$ macrophages was observed in Inflam$^{HIGH}$Vasc$^{LOW}$ (2.9%, 0.69–5.12%) as compared to Inflam$^{LOW}$Vasc$^{HIGH}$ (1.26%, −0.39–2.9%) and Inflam$^{HIGH}$Vasc$^{HIGH}$ (0.47%, 0.03–0.91%) groups (Fig. 4a). Additionally, flow-based immunophenotyping on 3 BCC tumors indicated higher activated CD8$^+$GzmB$^+$ and CD8$^+$ Ki67$^+$ cells in Inflam$^{HIGH}$Vasc$^{HIGH}$ as compared Inflam$^{LOW}$Vasc$^{HIGH}$, also suggesting the prevalence of inflamed state in the Inflam$^{HIGH}$ phenotype (Fig. S4b–d).

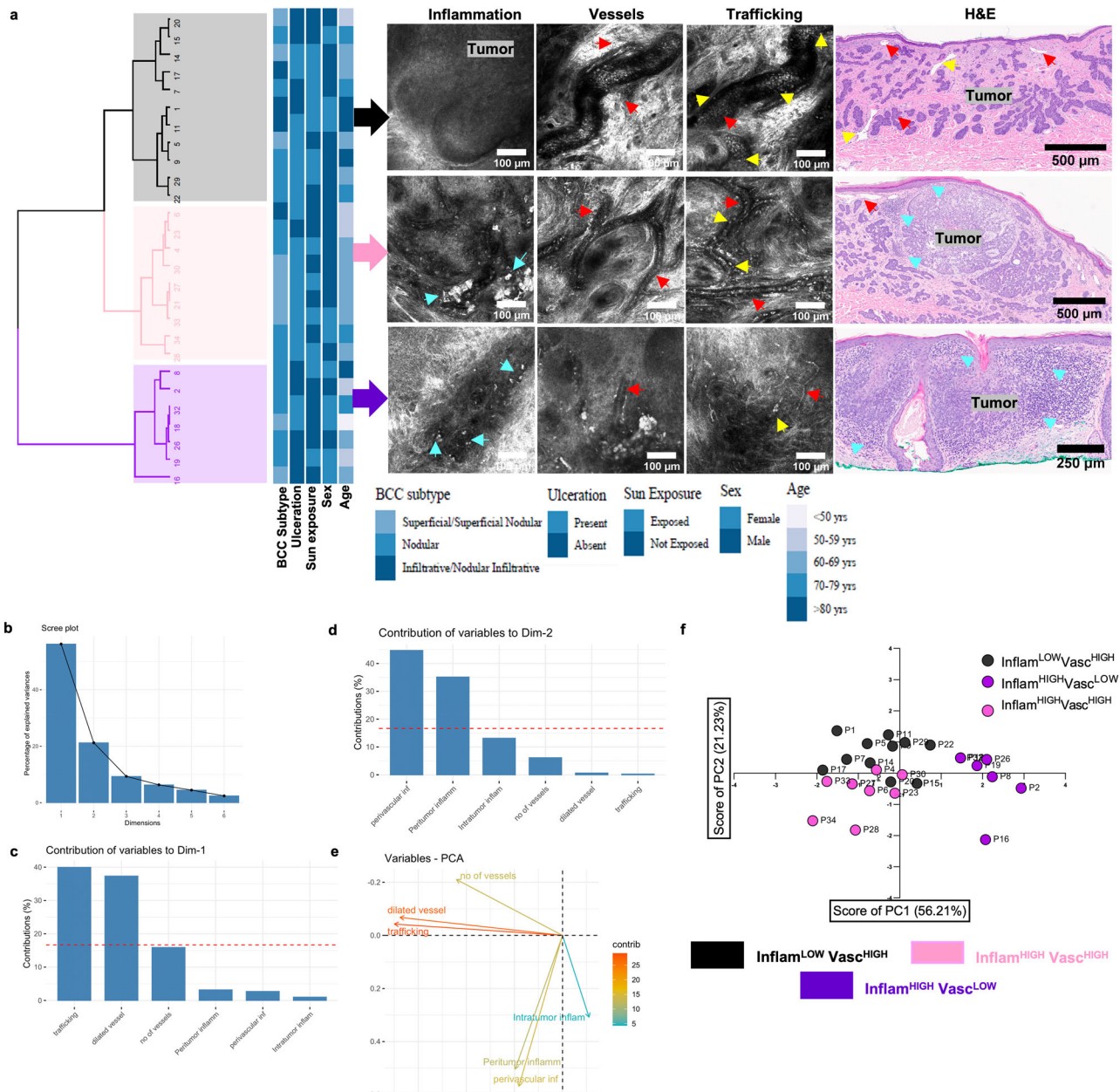

**Fig. 2 | Unsupervised clustering identifies three main RCM TiME phenotypes and assigns groups based on inflammation and vasculature. a** Unsupervised statistical clustering on major RCM features (inflammation, vasculature, trafficking) on n = 27 distinct BCCs yields 3 main phenotypes. No phenotypic association with any clinical features was observed. Representative RCM features within each phenotype and corresponding H&E are shown (cyan arrow-immune cells, yellow arrow-trafficking, red arrow-blood vessels, H&E scale bar- 500 μm). **b** Scree plot showing percentage contribution to variance for each PC. Top 2 PCs encompassing ~77% variance were used for elucidating phenotype nomenclature. **c** Vascular features-trafficking, dilated vessel and number of vessels- show predominant contribution in dimension 1 (PC1). **d** Inflammation features- perivascular, peritumor and intratumor inflammation mainly contribute to dimension 2 (PC2). **e** Contribution of variables to the PCs. **f** Scatter plot using contribution from PC1 and PC2 highlights 3 clusters in the principal component analysis (PCA). The phenotypes were assigned as Inflam^LOW Vasc^HIGH (black), Inflam^HIGH Vasc^LOW (purple) and Inflam^HIGH Vasc^HIGH (pink) since PC1 classifies phenotypes based on vascular features while PC2 classified phenotypes on the basis of inflammation. RCM images were selected after reviewing images in the entire dataset. The selected images are the most representative based on PC contribution within each group. Source data are provided as a Source Data file. RCM reflectance confocal microscopy, PCA principal component analysis, PC principal components, BCC basal cell carcinoma.

## Lymphocytes show strong association while tertiary lymphoid structures (TLS) show minimal association with TiME phenotypes

Tertiary lymphoid structures (TLS) are lymphoid formations that form in nonlymphoid tissues at sites of chronic inflammation and are associated with improved patient outcomes and response to immunotherapies[31,32]. We investigated TLS as markers for immune states. As seen in multiplexed immunofluorescence, large variation was observed in the T-cell and B-cell immune infiltrates and tertiary lymphoid structures (TLS) across patients. Total CD3+ and CD20+ lymphocytes proportions were 9.2% (12%, 7.7–16%) in Inflam^LOW-Vasc^HIGH, 25% (25%, 15–36%) in Inflam^HIGH Vasc^LOW and 14% (14%, 8.4-20%) in Inflam^HIGH Vasc^HIGH groups. TLS area positivity did not differ significantly across the phenotypes (Fig. 4b).

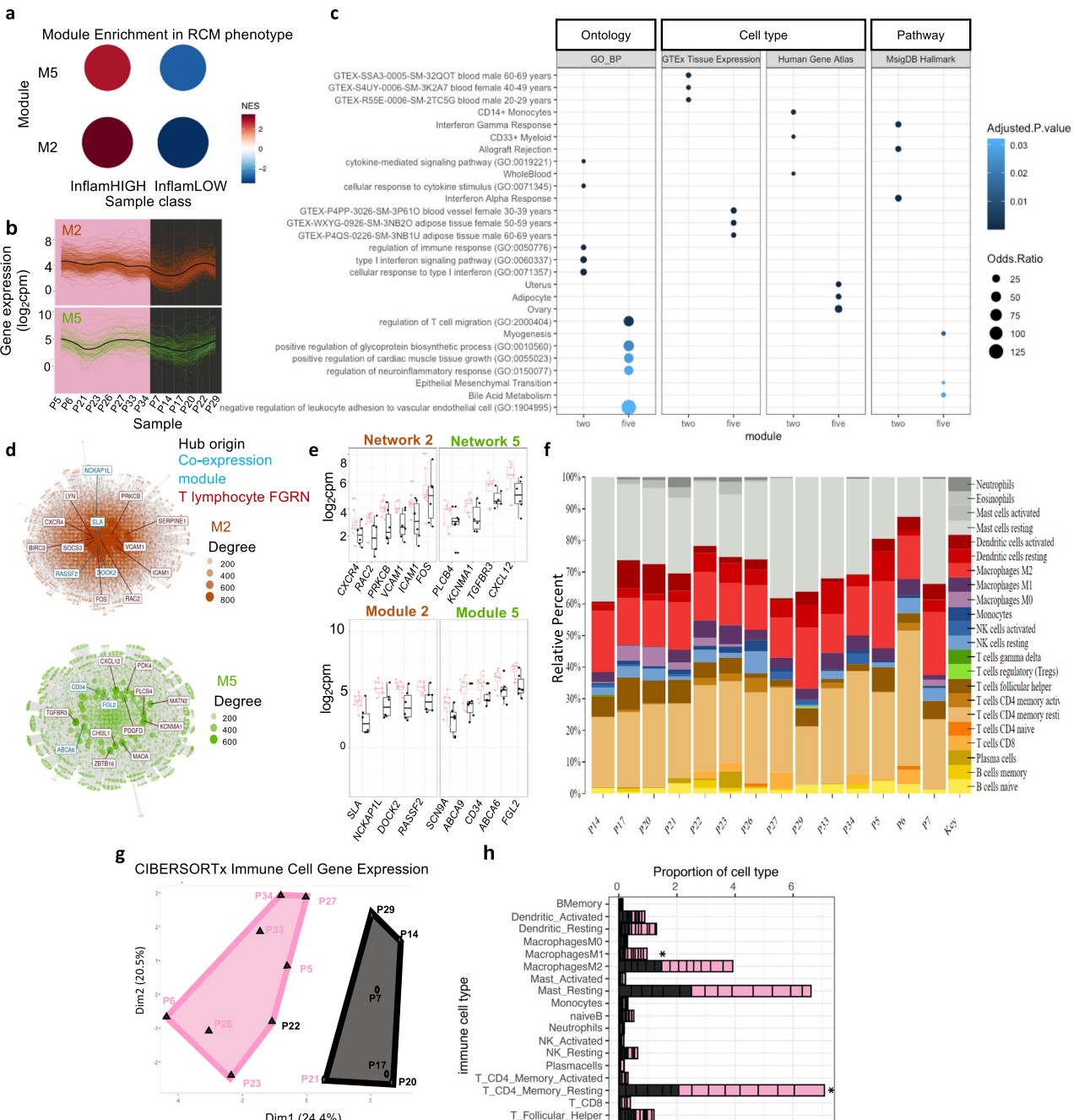

**Fig. 3 | Molecular signatures reveal inflammatory signatures predominantly correspond to Inflam[HIGH] phenotype. a** Significant enrichment of gene modules in RCM phenotypes from $n = 14$ BCC lesions (M2 NES = 3.5, adj. $p$ value = 0.00061; M5 NES = 2.9, adj.$p$value=0.00061). **b** Profile plots of genes in modules 2 and 5. Colored lines show expression levels for individual genes and the black line represents mean expression (log₂cpm) of all genes in the module. Individual samples are displayed on x axis and colored by RCM phenotype (high-inflammatory= pink, low-inflammatory = black). **c** Gene ontology (GO) enrichment of biological processes (BP) along with genes associated with cell type specificity (GTEx Tissue Expression and Human Gene Atlas) and cellular pathways (MsigDB Hallmark) are shown for modules 2 and 5 (adj.$p$value < 0.05 with multiple testing correction using BH; top 5 terms for ontology and top 3 for cell type and pathways when applicable). **d** Gene networks of M2 and M5 in T-lymphocytes. Top 10 most connected genes (Hub) in network are labeled (interaction = red). Module hub genes identified in network are indicated in blue (co-expression). Nodes indicate genes (size is proportional to degree) and edges represent connections to genes in network. **e** Box plots for

expression of network hub genes (top) and module hub genes (bottom). Individual points represent patient samples (pink = Inflam[HIGH], $n = 8$ biologically independent samples; black = Inflam[LOW], $n = 6$ biologically independent samples; differential expression was not significant at FDR < 0.05). Upper and lower whiskers extend from hinge to largest and smallest value no further than 1.5 * IQR. The lower and upper hinges correspond to 25th and 75th percentiles. Horizontal line represents median expression. **f** Relative proportions of the 22 immune cell types identified from CIBERSORTx in individual patients. **g** k-means clustering of transcript abundance in patient samples for genes assigned with immune cells deconvoluted from bulk RNA-seq with CIBERSORTx ($n = 547$ genes). **h** Relative proportions of individual immune cells in the Inflam[HIGH] (pink) and Inflam[LOW] (black) groups (individual patient samples indicated by black lines in bar chart) feature CD4[+] memory T-cells ($p$-value = 0.001) and M1 macrophages ($p$-value=0.012) as significant determinants of differences across the 2 groups using unpaired two-tailed Mann-Whitney test. Source data are provided as a Source Data file.

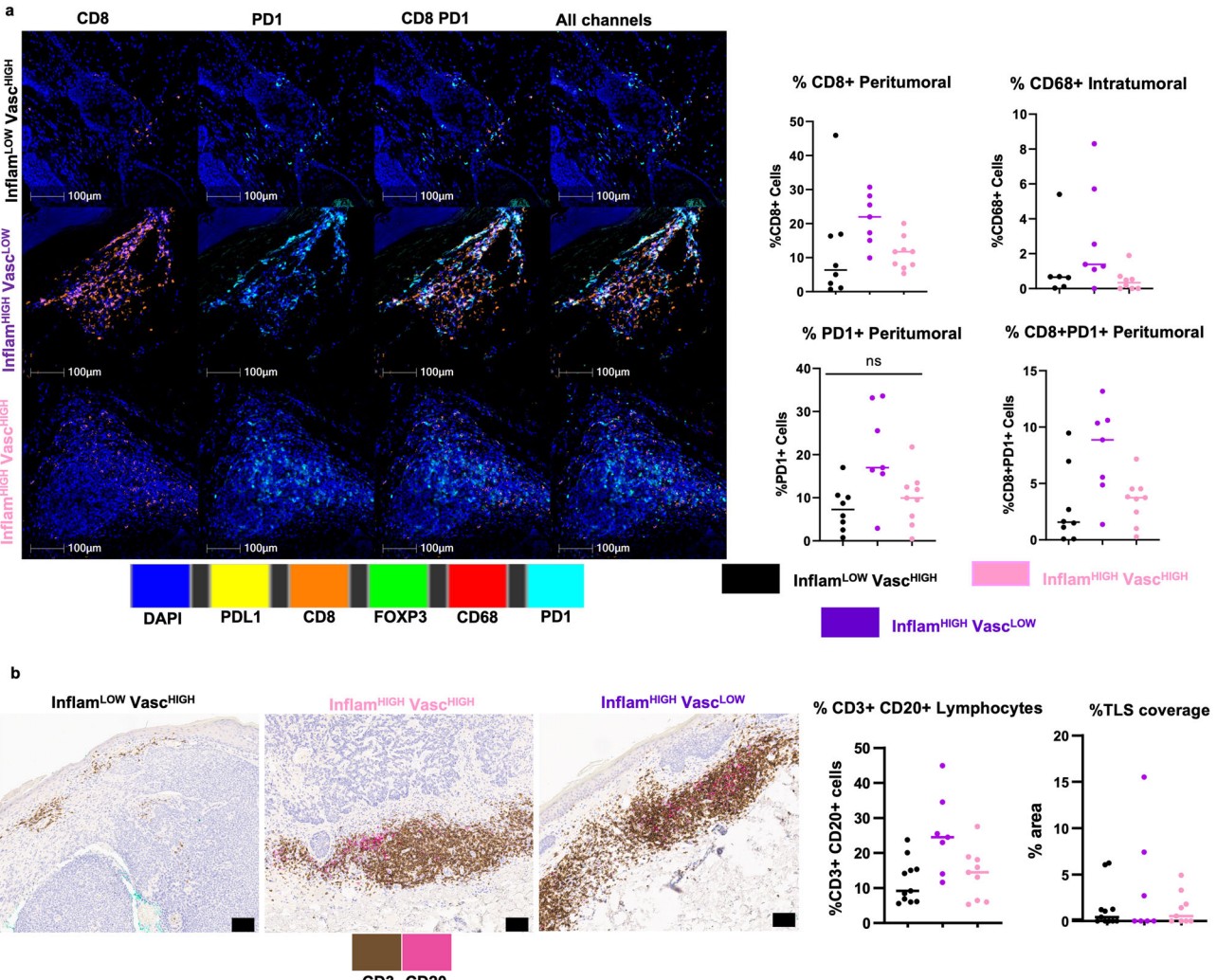

**Fig. 4 | Immunophenotyping through multiplexed staining correlates with RCM phenotypes. a** Representative images from multiplexed IF analysis (CD8⁺, FOXP3, CD68⁺, PD-1⁺ and PD-L1⁺) on $n = 24$ BCC specimens show presence of CD8⁺ T-cells, T-regs and macrophages in peritumoral infiltrates, along with PD-1 and PD-L1 expression in all three phenotypes: Inflam$^{LOW}$Vasc$^{HIGH}$ (black), Inflam$^{HIGH}$Vasc$^{LOW}$ (purple) and Inflam$^{HIGH}$Vasc$^{HIGH}$ (pink). Most abundant numbers of CD8⁺ cells ($p$-value= 0.031), PD1⁺ cells ($p$-value = 0.036), and highest fraction of CD8⁺ PD1⁺ cells ($p$-value= 0.030) was found in the Inflam$^{HIGH}$Vasc$^{LOW}$ phenotype, indicating an inflamed but exhausted phenotype. Distribution of CD68⁺ macrophages in intra-tumoral infiltrates was also highest in Inflam$^{HIGH}$Vasc$^{LOW}$ ($p$-value= 0.055). Data are presented as column scatter plots and median analyzed with Kruskal-Wallis test adjusted for multiple comparisons using Dunn's method. In peritumor analysis, $n = 8$, $n = 7$, $n = 9$ biologically independent specimens were analyzed from Inflam$^{LOW}$Vasc$^{HIGH}$, Inflam$^{HIGH}$Vasc$^{LOW}$ and Inflam$^{HIGH}$Vasc$^{HIGH}$ groups, respectively. In Intratumoral analysis, $n = 6$, $n = 7$, $n = 8$ biologically independent specimens were analyzed from Inflam$^{LOW}$Vasc$^{HIGH}$, Inflam$^{HIGH}$Vasc$^{LOW}$ and Inflam$^{HIGH}$Vasc$^{HIGH}$ groups, respectively. **b** Dual IHC staining for tertiary lymphoid structures using CD3⁺ T-cells (brown) and CD20⁺ B-cells (pink) in $n = 27$ BCC specimens demonstrate abundance in the Inflam$^{HIGH}$Vasc$^{LOW}$ and lowest values in the Inflam$^{LOW}$Vasc$^{HIGH}$ groups ($p$-value = 0.039). No clear phenotypic association with TLS coverage was found ($p$-value = 0.988). Data are presented as column scatter plots and median and median analyzed with Kruskal–Wallis test adjusted for multiple comparisons using Dunn's method. In this analysis, $n = 11$, $n = 7$, $n = 9$ biologically independent samples were analyzed from Inflam$^{LOW}$Vasc$^{HIGH}$, Inflam$^{HIGH}$Vasc$^{LOW}$ and Inflam$^{HIGH}$Vasc$^{HIGH}$ groups, respectively. Source data are provided as a Source Data file. IF: immuno-fluorescence; IHC: immunohistochemistry.

## Melanoma TiME phenotypes correlate with histopathology and T-cell infiltrates

Statistical clustering for 2 groups on the melanoma TiME features showed the presence of two main clusters: Inflam$^{LOW}$Vasc$^{HIGH}$ and Inflam$^{HIGH}$Vasc$^{HIGH}$. The phenotypic classes were assigned based on PCA, similar to BCC samples (Fig. 5a, Fig. S5a–e). No correlation of TiME phenotypes with tumor stage, age, gender or sun exposure was observed, although most of the invasive melanomas (superficial spreading subtype) belonged to the Inflam$^{HIGH}$Vasc$^{HIGH}$ phenotype. The CD3⁺ T-cell proportion was found to be 7.2% (7.2%, 5.2–9.2%) in Inflam$^{HIGH}$Vasc$^{HIGH}$ while 2.1% (2.2%, 0.49–3.9%) in the Inflam$^{LOW}$Vasc$^{HIGH}$ groups. As seen in BCC, TLS did not show a significant association with TiME phenotypes (Fig. 5b).

## Automated quantification of TiME features is feasible

Next, we investigated automated quantification of RCM TiME features – immune cells, leukocyte trafficking and vasculature, using machine learning and image processing algorithms. Quantification of immune cell density was explored using a U-Net segmentation model, which resulted in a Dice coefficient[33] (a common measure of accuracy) of 0.72 (Table 2). Representative images and segmentations are shown in Fig. S6a. Image-processing algorithms were used for quantification of vascular features (vessel area, diameter and number). Using manual vessel segmentation as ground truth, the Dice coefficient ranged between 0.29–0.78 (Table 2). Leukocyte trafficking was quantified using a custom pipeline involving image registration, background subtraction and particle tracking (detailed in Methods). High

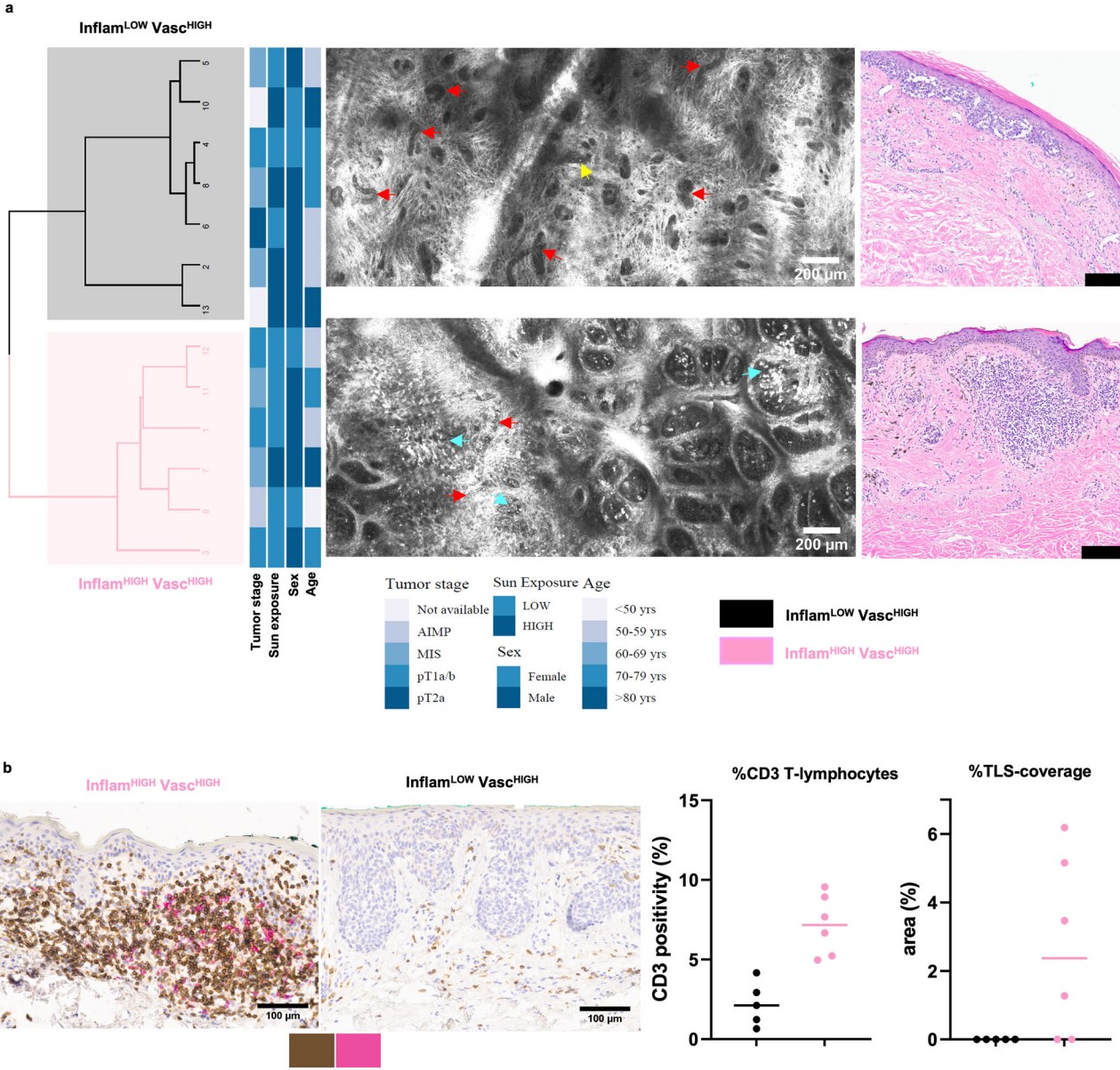

**Fig. 5 | Identical RCM TiME phenotypes in melanoma correlate with immune signatures. a** Unsupervised clustering of RCM features (inflammation, vasculature, trafficking) identifies two main phenotypes in melanoma lesions (*n* = 13) that are annotated as Inflam^HIGH^Vasc^HIGH^ and Inflam^LOW^Vasc^HIGH^ as shown in representative RCM and corresponding H&E images. (red arrows- vessels, cyan arrows- inflammation, yellow arrow-trafficking). **b** Dual IHC staining for tertiary lymphoid structures using CD3+ T-cells (brown) and CD20+ B-cells (pink) in melanoma specimens (*n* = 11) demonstrate higher abundance of CD3+ T-cells (*p*-value=0.004) and TLS (*p*-value =0.060) in the Inflam^HIGH^Vasc^HIGH^ group. Data are presented as column scatter plots and median analyzed with two-tailed unpaired Mann-Whitney test. In this analysis, *n* = 5 and *n* = 6 biologically independent samples were analyzed from Inflam^LOW^Vasc^HIGH^ and Inflam^HIGH^Vasc^HIGH^ groups, respectively. RCM images were selected after reviewing images in the entire dataset. The selected images are the most representative based on PC contribution within each group. Source data are provided as a Source Data file. RCM: reflectance confocal microscopy; IHC: immunohistochemistry.

agreement was observed between manual and automated counts at the optimization and final validation stages (Table 2). Parameters and representative examples for vessel and trafficking counts are shown in supplementary data (Fig. S6b–e). Subsequently, correlation of RCM TiME features with corresponding gene expression for inflammation, angiogenesis and trafficking suggested high correlation between total area % inflammation with myeloid cells, dendritic cells and T-cell markers (*CSF1R*, *CD1E* and *CD3E*), and total leukocytes-like area % with cytotoxic T-cell markers (*CD8B* and *GZMA*). Vascular features such as vessel diameter and trafficking were also correlated with angiogenesis and endothelial markers (*VEGFD*, *VEGFA* and *PDGFD*), and trafficking markers (*CCL-18*, *CAV-1* and *CCL28*), respectively (Fig. 6a). These TiME

features were additionally used as traits in module-trait analysis to determine relationships with gene co-expression modules shown to be enriched in the Inflam^HIGH^ groups (Fig. 3a). A significant correlation between M5 and total myeloid (dendritic cell, macrophages) inflammation area was discovered (Figs. 6b, S6f).

**Treatment response to imiquimod can be predicted by combined immune and vascular features and TiME phenotyping**
Treatment response to the TLRA immune-therapy in BCC patients was correlated with TiME features and phenotypes. Unsupervised statistical clustering yielded 2 groups: one group for responders (5 of the 7 responders) and the other for non-responders (4 of the 6 non-

**Table 2 | Automated quantification of RCM features is feasible**

| TiME feature | Test/validation data | Validation result |
|---|---|---|
| Inflammation (U-Net model) | 9.7% | Dice coefficient: 0.72 |
| Vessel segmentation | 5.2% | Dice coefficient: 0.29–0.78 |
| Trafficking – optimization | 10.9% | Spearman r: 0.79–0.82 |
| Trafficking – final validation | 2.5% | Spearman r: 0.74–0.89 |

Summary of results for validation for each automated quantification approach using data from n = 92 distinct lesions. Quantification of immune cells (leukocyte-like, dendritic cells and macrophages) using a 3-class UNet model resulted in Dice coefficient of 0.72. Segmentation of blood vessels for quantifying area and diameter of vessels showed a wide range of accuracy. Automated leukocyte trafficking counts were correlated with manual counts as ground truth during optimization and validation, demonstrate Spearman r between 0.74–0.89 depending on track length.

responders). The responder group was characterized mainly by high inflammation, while the non-responder group showed both high inflammation and vasculature (Fig. 7a). Phenotype-prediction of responders and non-responders was performed by overlaying on the original BCC phenotyping scatter plot (Fig. 2f). Most responders (5 of 7) belonged to the Inflam$^{HIGH}$Vasc$^{LOW}$ phenotype (Fig. 7b). Further, evaluation of differences in TiME features between responders and non-responders demonstrate increased number of vessels and stromal macrophages/dendritic cells in non-responders. Higher leukocyte trafficking, vessels, and stromal macrophages were present in 50%, 100%, and 86% of the non-responders, respectively. Although intra-tumoral inflammation and tumor infiltrating lymphocyte (TILs)-like features were similar, vessel density was found to be significantly different between responders and non-responders (Fig. 7c). Linear separability plots confirmed addition of vessels to inflammation enhanced separation between responders and non-responders (Fig. S7a). Linear regression models for response prediction (Fig. S7b) demonstrate low predictive power of inflammation as a variable, either as "TIL-like cells" or "intratumoral inflammation" with accuracy of 46% and 61%, respectively. Addition of stromal vessels to intratumoral inflammation or TIL-like cells as features in the linear regression model resulted in best model performance (71% sensitivity, 83% specificity and 76% accuracy) (Table 3).

## Discussion

Phenotyping the tumor microenvironment beyond simply T-cell infiltration is crucial to develop robust predictive platforms for patient stratification during immunotherapies. Since not all T-cell inflamed or hot phenotypes respond to treatment, tumors seemingly utilize additional mechanisms for evading immune response and establishing an immune-suppressive microenvironment. The tumor vasculature plays an important role in mediating this immune suppression and immune exclusion. To comprehensively evaluate the inflammation-vascular axis, we integrated features from inflammation and vasculature to investigate the presence of distinct in vivo TiME phenotypes in skin cancers that are predictive of patient response to therapy. The TiME features imaged non-invasively on RCM show high agreement with the gold standard histopathology, confirming the feasibility of detecting these features in vivo in patients (Table 1). Using a combination of vessel and immune features, we derived phenotypes using unsupervised clustering to minimize subjective bias. Within the BCC dataset, phenotyping using unsupervised clustering yielded three main phenotypes (Fig. 2a). In the smaller melanoma dataset, the clustering was programmed for two unsupervised clusters (Fig. 5a). While the Inflam$^{LOW}$Vasc$^{LOW}$ phenotype is a clinical possibility, skin cancers are typically highly immunogenic and vascular tumors, therefore we did not expect to encounter this phenotype in this smaller dataset[34–36]. Assessment of large cohorts would be needed to identify, and study the effect of this phenotype on treatment response in skin and other cancers. The TiME phenotypes in BCC and melanoma

strongly correlated with inflammatory molecular signatures, along with T-cell and macrophage abundance (Figs. 3, 4a, b, 5b). Furthermore, the phenotypes and combined vascular/immune features better correlated with treatment response than inflammation as a singular feature and best treatment response was observed in Inflam$^{HIGH}$Vasc$^{LOW}$ phenotype (Fig. 7a–c, Table 3), highlighting the importance and potential clinical utility of this approach in integrating both inflammation and vasculature to characterize and phenotype TiME.

Specifically, analysis of gene co-expression modules generated from all variable genes expressed across 14 BCC samples successfully identified eight modules of co-regulated genes, of which 2 modules were significantly enriched in RCM phenotypes (Fig. 3a, b). The modules show distinct regulation of immune cell and vascular function where M2 was enriched for genes associated with T-cell activation and differentiation, myeloid cell differentiation and leukocyte adhesion, in contrast to M5 enrichment of genes involved in blood vessel proliferation and cell adhesion (Fig. 3c). Given the shared concentration of genes participating in immune cell and vascular functions for both M2 and M5, albeit potentially from distinct cell origins, we generated a gene interaction network using gene connections in T-lymphocytes, macrophages, blood and skin. Together, M2 and M5 T-lymphocyte networks appear to engage in cell type specific pathways in agreement with module enrichment in biological processes similarly describing contents of blood, i.e. immune cells, for M2 whereas cell types comprising the blood vessel were enriched for M5.

Investigations into specific genes and pathways within the module and network hub genes demonstrate higher prevalence of genes crucial for immune cell migration (including leukocyte trafficking) and vascular function such as *ICAM1, VCAM1, CXCR4, CXCL12, NCKAPL1, DOCK2, PDGFD* and *TGFBR3* in Inflam$^{HIGH}$Vasc$^{HIGH}$ phenotype (Fig. 3d, e). This suggests this phenotype likely had comparatively 'normal' vasculature[37]. Conversely, Inflam$^{LOW}$Vasc$^{HIGH}$ show downregulation of major adhesion molecules (*ICAM1, VCAM1*) along with significantly lower immune cell signatures (Fig. 3d, e), suggesting features of endothelial anergy and anergy-induced immune excluded state[16,37]. This immune-exclusion and endothelial anergy state may have been potentiated by immunosuppressive tumor-intrinsic factors (*CTNNB1, PTEN, COX11*)[38,39] in the Inflam$^{LOW}$Vasc$^{HIGH}$ phenotype, which needs to be studied in context of tumor genetic signatures, mutational burden and immune exclusion[40–42].

Furthermore, the majority of module hub genes (80% of the top 5 genes in M2 and M5) were found to have known immune functions (Fig. 3d-e). Notably, network hub genes are largely shared across cell types, regardless of tissue origin of gene network. This exemplifies the robustness of the M2 and M5 network hub genes, through participation in shared pathways across functionally distinct tissues and contributing to observed differences in RCM phenotypes. Most of the network hub genes common across gene networks are also involved in inflammatory or vascular pathways. Few module and network hub genes *(SCN9A, PLCB4, RASSF2)* have minimal or no direct immunological function, although they show connectivity within gene networks enriched in skin, macrophage, blood and T-lymphocyte pathways. While *SCN9A* encodes the voltage-gated ion channel $Na_v$ 1.7 and is primarily associated with pain disorders, it can also influence chemokine-induced migration of the CD1a$^+$ dendritic cells[43]. Similarly, *PLCB4* or phospholipase C beta-4 enzyme has established neurological roles, and was recently shown to play a role in selective promotion of CD8-T cell-dependent adaptive immunity[44,45]. Further, *RASSF2*, a known tumor suppressor gene that promotes apoptosis and cell-cycle arrest is also implicated in suppression of immune responses, angiogenesis, and metastasis[46]. Enrichment of these genes in the Inflam$^{HIGH}$Vasc$^{HIGH}$ phenotype draws parallels to these studies and strongly suggests previously unknown roles of these relatively non-immune genes in regulating inflammation and/or vasculature within the tumor microenvironment. Although the DEG analysis on bulk RNA-seq data

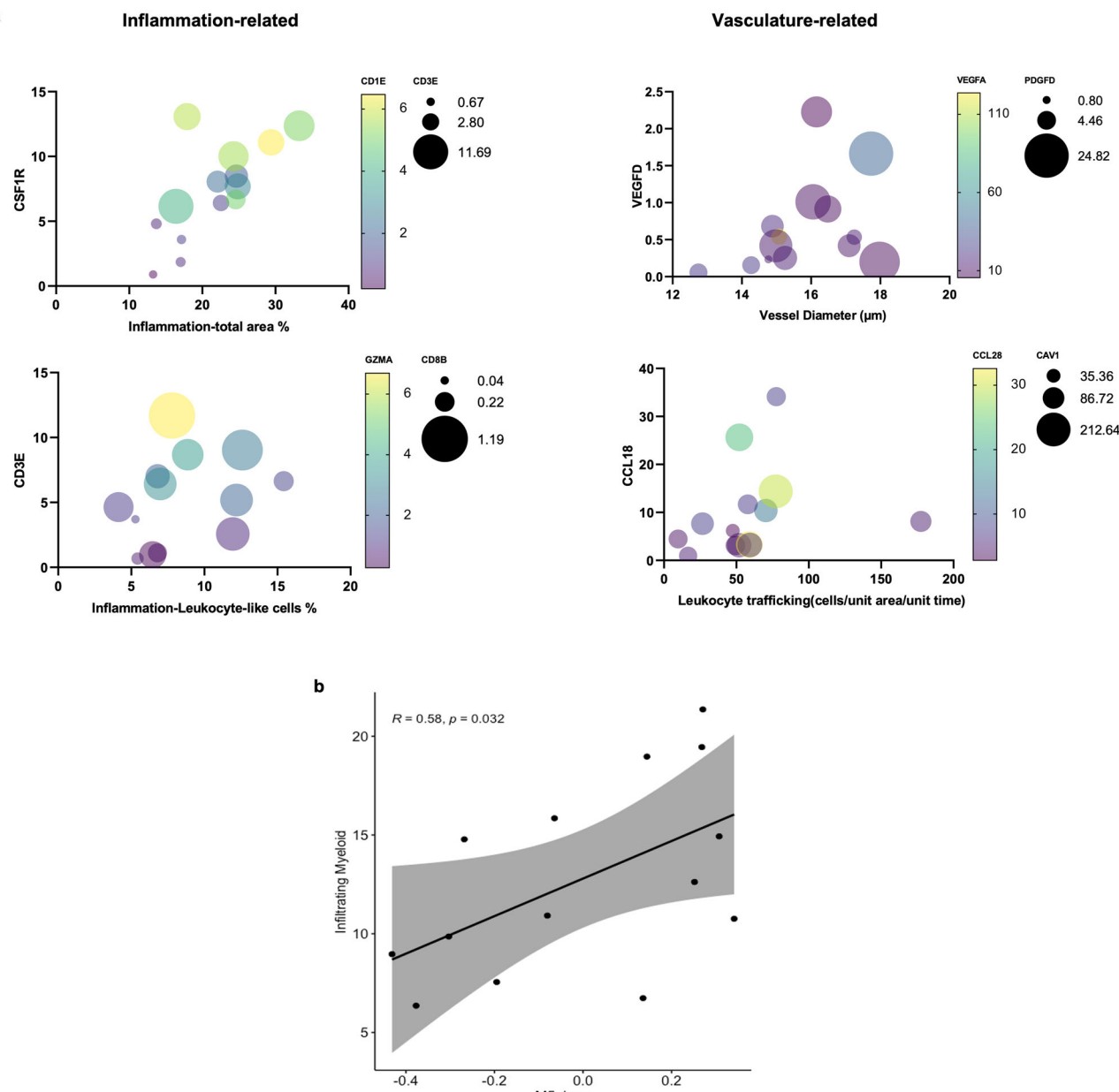

**Fig. 6 | Quantified RCM TiME features correlate with gene expression.**
**a** Automated features correlated with corresponding gene expression in $n = 14$ BCC lesions shows moderate to high correlation between total inflammation area with *CSF1R* (r = 0.73, CI: 0.32 to 0.91, *p*-value = 0.002), *CD1E* (r = 0.64, CI: 0.15 to 0.87, *p*-value= 0.008) and *CD3E* (r = 0.51, CI:−0.04 to 0.82, *p*-value = 0.032) and between total leukocyte-like cells area with *CD3E* (r = 0.6, CI: −0.13 to 0.79, *p*-value = 0.013), *CD8B* (r = 0.6, CI: 0.1 to 0.86, *p*-value = 0.012) and *GZMA* (r = 0.53, CI: −0.01 to 0.83, *p*-value = 0.026). Similarly, vessel diameter and leukocyte trafficking were correlated with *VEGFD* (r = 0.459, CI: −0.1 to 0.80, *p*-value = 0.050), *VEGFA* (r = −0.477, CI: −0.81 to 0.09, *p*-value = 0.044), *PDGFD* (r = 0.538, CI: 0 to 0.84, *p*-value = 0.025), and trafficking with *CCL18* (r = 0.561, CI: 0.019 to 0.84, *p*-value= 0.042), *CAV-1* (r = 0.468,

CI: −0.10 to 0.80, *p*-value= 0.046) and *CCL28* (r = −0.42, CI: −.016 to 0.78, *p*-value= 0.137), respectively. Non-parametric two-tailed Spearman correlation was computed across each dataset. **b** Automated features correlated with gene co-expression modules show total myeloid cells on RCM (dendritic cells+macrophages) were significantly correlated with eigengene values for M5 module(Spearman method, *p*-values estimated using *t*-distributions). Confidence interval at 95% indicated in gray along with linear regression line and correlation coefficient (*p*-value = 0.032). Source data are provided as a Source Data file. *CSF1R*: colony stimulating factor 1-receptor; *CD*: cluster of differentiation; *GZMA*: gran-zyme A; *VEGF*: vascular endothelial growth factor; *PDGFD*: platelet derived growth factor D; *CCL*: CC-chemokine ligand; *CAV*: caveolin.

did not identify other major immunological or vascular genes, attrib-uted to the low sample size and high variability within patient data, our analysis into co-expression of variable genes enabled detection of known and previously unknown genes important in resolving phenotypes.

Significantly, clustering on the CIBERSORTx data which enriches for immune cells genes from 22 cell types explained assignment of samples to RCM phenotypes (Fig. 3g). CIBERSORTx also identified major differences in T-cells and macrophages across the two groups,

which were subsequently validated using immunostaining (Figs. 3f, h and 4). While heterogeneity in immune cell populations through CIBERSORTx does resolve RCM phenotypes, it only captures one component i.e. inflammation, within the phenotype. We anticipate that similar heterogeneity in cell populations comprising the stroma such as fibroblasts and endothelial cells will also capture distinguishing features between these phenotypes, these were not addressed in this study. Since genes involved in immune response regulation and function yet possibly originating from non-immune cells featured

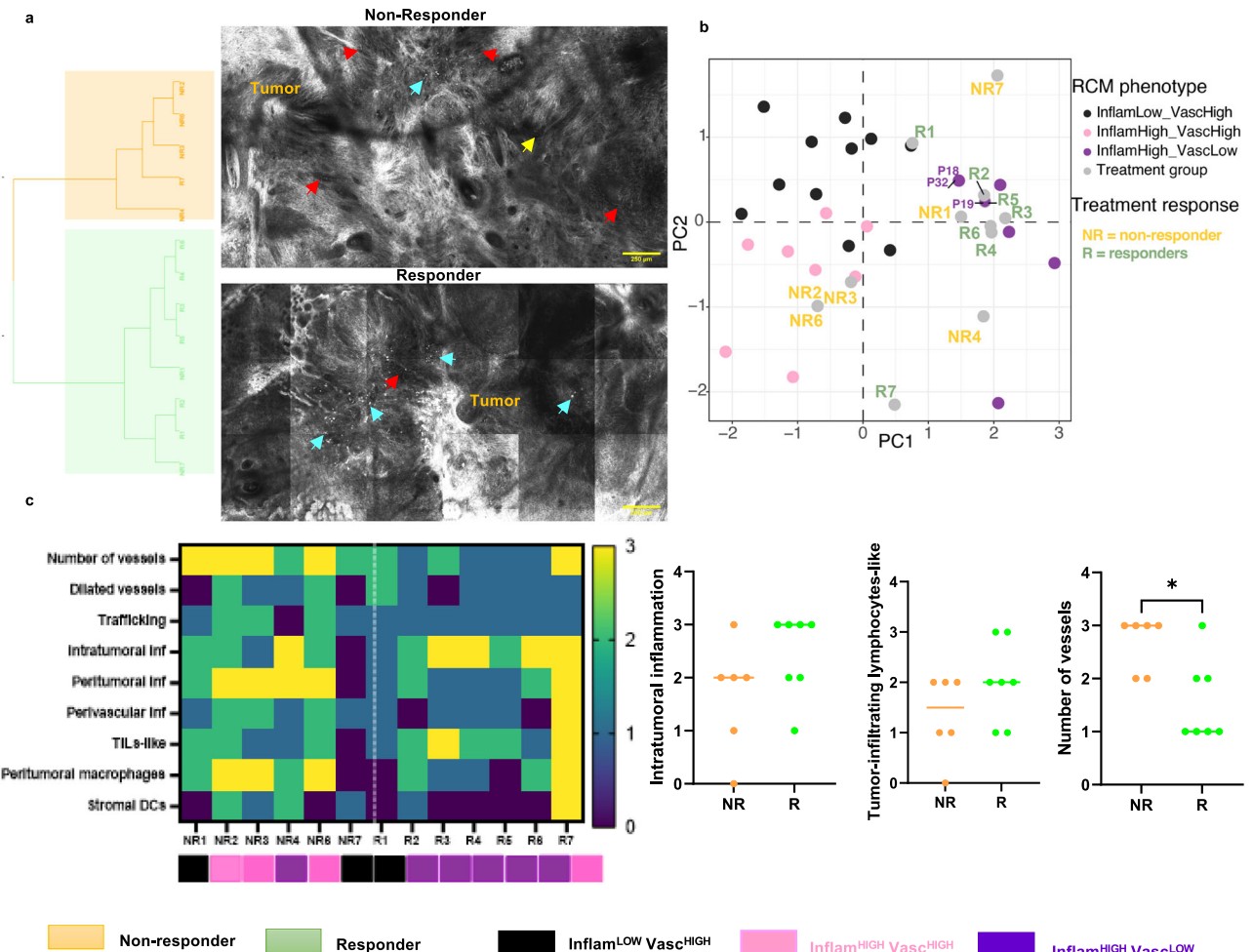

**Fig. 7 | Inflam^HIGH^Vasc^LOW^ phenotype corresponds to highest response in TLRA treated patients. a** Unsupervised statistical clustering on $n = 13$ BCC lesions receiving TLRA treatment yields two major clusters of mainly non-responders (orange) and responders (green), attributed to the presence of differential TiME features shown in representative RCM images (red arrows- vessels, cyan arrows-inflammation, yellow arrow-trafficking). **b** TiME phenotyping of TLRA patients predicted by overlaying on the original BCC phenotype PCA plot (Fig. 2b) suggest that most responders (R) belonged to the Inflam^HIGH^Vasc^LOW^ group. **c** Comparison of major TiME features across responders ($n = 7$) and non-responders ($n = 6$) demonstrate inflammation features (intratumoral inflammation, TILs-like cells) are insufficient to stratify patients based on response. However, stromal vessels can differentiate between responders and non-responders ($p$-value= 0.035). Data are presented as column scatter plots and median analyzed using two-tailed Mann-Whitney test. RCM images were selected after reviewing images in the entire dataset. Selection was based on the features most prominently observed within each class. Source data are provided as a Source Data file. TLRA: toll-like receptor agonist, BCC: basal cell carcinoma, PCA: principal component analysis; TiME: tumor-immune microenvironment; RCM: reflectance confocal microscopy.

### Table 3 | Modeling of vascular features with immune cells improves response prediction accuracy

| Features | Sensitivity | Specificity | Accuracy |
|---|---|---|---|
| Intratumoral inflammation | 0.51 | 0.33 | 0.46 |
| Tumor-infiltrating lymphocytes-like | 0.71 | 0.5 | 0.61 |
| Intratumoral inflammation + number of vessels | 0.71 | 0.83 | 0.76 |
| Tumor-infiltrating lymphocytes-like + number of vessels | 0.71 | 0.83 | 0.76 |

Linear regression modeling of RCM features for response prediction using Akaike information criterion demonstrate improved classification accuracy by combining immune and vascular features, as opposed to inflammation alone.

prominently in the module and network hub genes, future studies should also address cellular heterogeneity in these stromal cell populations using unbiased approaches such as single-cell RNA-sequencing that may comprehensively address the source of gene expression differences across RCM phenotypes.

Using immunophenotyping through dual IHC and multiplexed IF, the RCM phenotypes correlated with peritumoral abundance of CD3^+^, CD20^+^, CD8^+^ and CD8^+^ PD-1^+^ T-cells in BCC, and CD3^+^ T-cells in melanoma (Fig. 4a, b and 5b). Similar trends for abundance of myeloid cells

(macrophages) and regulatory T-cells were found in the Inflam^HIGH^ as compared to the Inflam^LOW^. While the strongest difference in CD8^+^ cells was found between Inflam^HIGH^Vasc^HIGH^ and Inflam^HIGH^Vasc^LOW^ groups, the overall CD8^+^ cells in the Inflam^LOW^Vasc^HIGH^ group were lowest. One outlier within this group had intense immune infiltrates in the deeper dermis, explaining very high lymphocytic density on IHC which was not found in RCM. Furthermore, tentatively excluding this outlier patient from Inflam^LOW^Vasc^HIGH^ led to a significant difference in CD8^+^ T-cells between Inflam^HIGH^ and Inflam^LOW^ groups. No association was

observed with the area of tertiary lymphoid structures, (TLS) a hallmark of an inflamed micro-environment, and positive cancer outcomes[32], in both melanoma and BCC. We suspect this discrepancy to be due to most of the TLS arising much deeper in the tissue, outside the field-of-view of RCM (depth-limited to 0.25 mm). In conclusion, a comprehensive immunohistochemical assessment using multiplexed IHC supports the described distinct phenotypes as identified through RCM. These results were also corroborated on the flow-based immunophenotyping on 3 patients that showed higher activated CD8+ cells in the Inflam[HIGH]Vasc[HIGH] phenotype (Fig. S4b–d).

Although immune-based therapies have shown unprecedented and durable responses, the response rates have been variable. Topical TLRA imiquimod immune-therapy has shown response rates between 60-84% in superficial melanoma and basal cell carcinoma[47,48], highlighting the need to identify the responders and non-responders early to minimize treatment side-effects and patient morbidity, and streamline management. Ulceration, *LINC* and PD-L1 expression, and perifollicular infiltration of melanocytes have shown some association with response to imiquimod in BCC and superficial melanoma[49–51], but are not often used for informing treatment decisions. There is a need for more comprehensive and quantitative analysis of major determinants of anti-tumor immunity within TiME that enable patient stratification at the bedside. Because of the dynamic nature of the interactions within TiME, in vivo imaging is crucial for studying dynamic processes, including active vascular processes such as leukocyte trafficking which is optimally studied as live dynamic events, as opposed to ex vivo tissue studies on vasculature which can show inconsistent vessel measurements[20]. As skin cancers are accessible for such non-invasive imaging, we evaluated thirteen patients undergoing prospective treatment with topical TLRA. Seven patients responded to treatment (complete tumor clearance) while 6 did not respond (partial or no tumor clearance). On this small cohort, the Inflam[HIGH]Vasc[LOW] phenotype correlated with maximal response, and inclusion of vascular features in predictive models led to improved response prediction accuracy (Fig. 7c, Table 3). On RCM, this phenotype demonstrated lower immunosuppressive angiogenic features with higher intratumoral inflammation. This phenotype also presented as the inflamed exhausted phenotype on multiplexed IF with highest abundance of CD8+ and CD8+ PD-1+ T-cells (Fig. 4a). Notably, only Inflam[HIGH]Vasc[LOW] showed highest response to the TLRA immune-therapy (Fig. 7b). Therefore, we speculate that the Vasc[HIGH] phenotypes could benefit from adjunct vascular-targeted therapies in combination with immune-modulating therapies. However, specific mechanisms of treatment resistance attributed to "vasculature" will first need to be resolved for the two groups within Vasc[HIGH] by studying TLRA response on a large patient cohort. For example, vessel normalization through pharmacological targeting of the Wnt/β-catenin pathway and anti-angiogenic topical treatments (COX-2, basic fibroblast growth factor or bFGF inhibitors) can overcome endothelial cell anergy and, reinduce/enhance adhesion molecule expression for increased leukocyte infiltration of effector T cells into tumors[15], especially benefiting Inflam[LOW]Vasc[HIGH] [52,53]. The predictive accuracy of 76% achieved by incorporating both inflammation and vascular features in our models (Table 3) can potentially be improved by incorporating adjunct stromal features like collagen and mucin in future studies, since these can also influence immune infiltration into tumors. Thus, this study confirms the feasibility of a predictive platform that will ultimately introduce a more individualized or personalized immune-therapy treatment approach, similar to the IMPACT[TM] Panel[54].

Using RCM and skin cancer as a model, we demonstrate the proof-of-concept for unperturbed characterization and phenotyping of TiME inside patients and the feasibility to predict response/no-response before treatment. However, manual evaluation of TiME features would require an effort that is unrealistic in clinical practice. Manual approaches would also inevitably involve extensive training of readers

in feature recognition. The current era of machine learning and artificial intelligence allows extraction of quantitative biomarkers from images to inform on disease characterization, monitoring and assessment of response to treatment, also a mission of the quantitative imaging network for radiological imaging[55,56]. Quantitation also has the potential to minimize intra- and inter-observer variability of feature evaluation and provide objective decision-support tools in patient management. Thus, towards real-world clinical implementation, we produced preliminary algorithmic pipelines for automated quantification of immune cell density, leukocyte trafficking frequency, and vessel thickness and density. The preliminary machine learning model trained on 2 subclasses of immune cells (leukocyte-like, and myeloid cells that included macrophages and dendritic cells) showed a 72% classification accuracy for estimating areas covered by immune cells (Table 2). Subsequent efforts will be directed towards instance segmentation (e.g., object detection) to estimate the occurrence statistics of cells, and the training will be extended to individual morphologies as shown in Fig. S1a. Through RCM imaging of blood vessels in humans, we uncovered important limitations that affect the accuracy, including difficulty in acquiring good quality data at specific anatomical locations and axial motion during data acquisition by the hand-held probe due to lack of stability. These limitations resulted in a more variable Dice coefficient range (Table 2). We have focused our efforts on overcoming these limitations by employing tissue coupling during video acquisition and using advanced frame stabilization tools to further improve the processing of vascular features. The quantified TiME features also demonstrate strong quantitative correlations with the amount of gene expression, suggesting the quantifications truly represented underlying immune or vascular features (Fig. 6a). M5 module genes which were enriched in the Inflam[HIGH]Vasc[HIGH] group also correlated with density of myeloid cells (macrophages, dendritic cells) (Fig. 6b), similar patterns were also seen in the multiplexed IF results (Fig. 4a). In this study, the quantified trafficking included all major trafficking events (rolling, adhesion, and crawling) (Supplementary Movies 2, 3) that correlated with *CCL18* and *CCL28* expression (Fig. 6a). Since specific genes orchestrate discrete steps within the trafficking cascade[57], better gene correlation with individual trafficking steps (e.g. *ICAM1/VCAM1* with rolling cells) can be expected. Thus, subsequent efforts will be focused on quantifying individual trafficking events.

Critically, through automated analysis, it is also possible to quantify longitudinal changes to monitor treatment-induced alterations e.g. normalized vasculature after vascular normalizing therapies could be analyzed in real-time to help assess response and uncover mechanisms of treatment resistance. This real-time longitudinal evaluation and quantification will contribute to treatment optimization and personalization of immune-therapy regimens. Similar to our recent efforts in automating image acquisition and diagnosis[58,59], in future, the quantification pipelines would be implemented on clinical devices to enable real-time quantification and TiME phenotyping for response prediction using validated quantitative predictive platforms. Thus, this may enable more precise and real-time clinical decision-making for patient stratification at the bedside.

Our study demonstrates a combination of high-resolution spatially resolved and dynamic imaging to advance current limitation in static detection of TiME features. Outstanding limitations of the approach include grayscale-limited specificity tissue contrast and imaging depth to 0.2–0.25 mm. The label-free approach enables visualization of all TiME features, but is limited in specificity for functional phenotyping. One major advantage is the feasibility of longitudinally monitoring spatio-temporal changes in immune cells and vasculature during treatment, akin to on-treatment biopsies. With the present state of RCM devices and technology, this approach is currently restricted to accessible diseases and cancers on the skin and mucosa. Complementary multimodal approaches[60] such as dynamic optical coherence tomography or optical frequency domain imaging

for imaging vasculature, lymphatics and tissue viability[61,62], multiphoton microscopy for better contrast and collagen delineation[63–65], photoacoustic microscopy for functional vascular imaging[66] and fluorescence lifetime imaging for probing immune cell specificity and activation states[67] will further enhance in vivo TiME visualization and enhance current TiME phenotyping in the future.

In subsequent studies, extensive validation with targeted molecular analyses on precision biopsies[68] will enable better correlations. Since only the most prominent signals can be detected in bulk-sequencing, other minor or rarer cellular differences in immune and stromal cells, especially in non-infiltrated samples may be detected using single cell or spatial transcriptomic analysis, to further our understanding of RCM phenotyping[69,70]. Through robust prospective studies on large cohorts, fundamental basis of phenotyping and their correlation with variable treatment responses in cancer immunotherapy systems can be explored for better patient stratification. This research could impact several oncological settings, including but not limited to cutaneous and mucosal, primary and metastatic keratinocyte and melanocyte cancers (melanoma, squamous cell carcinoma, basal cell carcinoma), and cutaneous lymphomas. The advances in TiME phenotyping presented here can enable hypothesis-driven research for developing new druggable targets and response predictive platforms, and gaining mechanistic insights on host anti-tumor immune response in cancers.

## Methods

### Study design
The study was approved by the Institutional Review Board at Memorial Sloan Kettering Cancer Center (MSKCC-IRB). Written informed consent was obtained from all participants recruited under active IRB protocols IRB#99-099 and IRB#21-019 included under the clinical trial NCT00588315. All research was performed strictly in accordance with the Declaration of Helsinki and relevant guidelines and regulations. The main research objectives for this study were as follows: (i) investigate TiME phenotyping in skin cancer patients, (ii) investigate biology and immunological states underlying TiME phenotyping, (iii) correlate specific TiME features and phenotypes with response to a toll-like receptor agonist (TLRA) therapy. This observational cross-sectional study involved RCM imaging on human patients presenting with skin lesions referred for dermatology consultation in the Dermatology Service at Memorial Sloan Kettering Cancer Center (MSKCC), New York. Patients (aged 18 or older) with either a previously biopsied or clinically suspected skin cancer or drug rash amenable for imaging were prospectively enrolled between November 2018 and November 2021. On a subset of patients, patients undergoing topical TLRA treatment (5x/week for 6 weeks) as standard of care were imaged at baseline before treatment (T0), and end of treatment to confirm tumor clearance. Validation of phenotyping was investigated on tissues obtained as part of standard of care or research by histopathology, bulk RNA-sequencing, multiplexed IF and dual IHC to correlate with underlying immune and vascular features. Automated quantification of TiME features, and modeling of treatment response on TiME features and phenotypes was performed for near-future clinical applications in predicting treatment responses (Fig. 1).

### In vivo imaging
In vivo RCM imaging was performed prospectively on 118 lesions using RCM (VivaScope 1500 or handheld VivaScope 3000, (Caliber Imaging and Diagnostics, Rochester, NY, USA) and/or an integrated handheld RCM-OCT prototype. VivaScan v10.0 (Caliber Imaging and Diagnostics, Rochester, NY, USA) was used to acquire images and images were interpreted in real-time at the bedside to select representative areas with tumor, immune cells and blood vessels across the lesion by 2 investigators (MC and AS) having more than 5 years of RCM reading experience. Surrounding normal skin was also imaged as control, but

excluded from analysis since it lacked tumor and TiME features. Mosaics (large area sampling), stacks (depth sampling), scanning and single field-of-view (FOV) videos were acquired from multiple regions within the lesion and saved in an online database (Vivanet, Caliber ID, Rochester, NY) or on a local drive. Patients visiting Dermatology service were recruited in this study. Twenty-eight BCC lesions from 20 patients (9 females, 11 males; age range 43–88 years) were used for the BCC phenotyping analysis. Thirteen melanoma lesions from 12 patients (4 females, 8 males; age range 46–87 years) were used for melanoma phenotyping. Thirteen lesions on 8 patients (4 females, 4 males; age range 54–70 years) undergoing TLRA agonist (imiquimod) treatment were included in this study. Imaging was performed before treatment or standard of care biopsy for all patients. Imaging was performed once for a given lesion and not repeated due to clinical constraints. In the imiquimod dataset, patients were imaged after treatment to investigate tumor clearance.

### Patient tissue
Biopsies (targeted or non-targeted) taken as standard of care or for research use were used for histopathological, immunohistochemical, RNA-sequencing and flow cytometry correlations. Formalin-fixed paraffin embedded (FFPE) specimens from 39 lesions (27- basal cell carcinoma, 11- melanoma, 1-lichen-planus like keratosis) were used for histopathological and immunohistochemical (IHC) correlations. RNA-extraction was performed on 25 FFPE specimens with adequate tissue. However, RNA-seq was performed only on 14 out of 25 specimens representing the two groups Inflam$^{HIGH}$Vasc$^{HIGH}$ and Inflam$^{LOW}$Vasc$^{HIGH}$ due to limited RNA quantity. Multiplexed IF was performed on 27 FFPE BCC specimens, 3 specimens were excluded from the analysis since the tissue quality was heavily deteriorated during multiplexed staining. Imaging-guided small (2–3 mm) targeted biopsies[71] was performed on 5 lesions for frozen section histopathology and IHC ($n = 2$), and flow cytometry ($n = 3$). Although flow cytometry provides more quantitative estimates of cellular populations, acquiring adequate research specimens ($\geq 3$ mm) can potentially compromise patient care in a diagnostic setting and is not routinely practiced. Since we anticipated challenges in acquiring additional fresh tissue for flow cytometry beyond the first few samples, we concentrated our efforts on multiplexed IF and dual IHC on archived pathology specimens to validate the phenotypes.

### Manual RCM and histopathology evaluation
RCM features were manually evaluated (Fig. S1c, Table 1) by either of the 4 readers with at least 5 years' experience (AS) or >20 year experience in interpreting RCM images (MC, SG, CMAF). The major features evaluated on manual reading included number of vessels, dilated vessels, trafficking, intratumoral inflammation, peritumor inflammation and perivascular inflammation. These features were graded on a scale of 0–3 after exhaustive review of data from each patient. Data from 1 patient was excluded due to motion blur during image acquisition. For melanoma, specific features such as lichenoid inflammation, total lymphocyte-like cells were also evaluated along with spatial distribution of inflammation, immune cells and vasculature. For imiquimod response study, spatial distribution of vessels and type of three immune morphologies (dendritic cells, lymphocyte-like cells and macrophages), tumor-infiltrating lymphocyte-like cells, mucin and tumor regressing areas were also noted for more comprehensive assessment and correlation with treatment response. Same TiME features evaluated on RCM were graded on digitized histopathological slides by board-certified dermatopathologists (MG, MP). Manually evaluated features were used for statistical clustering for phenotyping and response prediction.

### Statistical clustering for TiME phenotyping
Unsupervised statistical clustering on manually evaluated TiME features was performed to explore classification trends or phenotypes

using HCPC and PCA in R using FactoMineR[72], FactoExtra[73] and ggplot2[74] packages. HCPC was performed using the HCPC function while PCA was performed using the prcomp function in R. The dendrograms were plotted using the fviz_dend functions. For PCA, the percentage contribution scree plot, the variable contribution and the biplot were obtained using fviz_contrib, fviz_pca_var and the fviz_pca_biplot functions, respectively.

## RNA extraction

FFPE sections from $n = 25$ specimens were deparaffinized using the mineral oil method. Briefly, 800 μL mineral oil was mixed with the sections and the sample was incubated at 65 °C for 10 min. Phases were separated by centrifugation in 360 μL Buffer PKD and Proteinase K was added for digestion. After a three-step incubation (65 °C for 45', 80 °C for 15', 65 °C for 30') and additional centrifugation, the aqueous phase containing RNA was removed and DNase treated. The RNA was then extracted using the RNeasy FFPE Kit (QIAGEN catalog # 73504) on the QIAcube Connect (Qiagen) according to the manufacturer's protocol with 285 μL input. Samples were eluted in 13 μL RNase-free water.

## Transcriptome sequencing

After RiboGreen quantification and quality control by Agilent BioAnalyzer, 356–500 ng of total RNA with DV200% varying from 88-93 underwent ribosomal depletion and library preparation using the TruSeq Stranded Total RNA LT Kit (Illumina catalog # RS-122-1202) according to instructions provided by the manufacturer with 8 cycles of PCR. Samples were barcoded and run on a HiSeq 4000 in a PE100 run, using the HiSeq 3000/4000 SBS Kit (Illumina). On average, 78 million paired reads were generated per sample and 20% of the data mapped to mRNA. While RNA extraction was performed on 25 FFPE specimens, due to limited RNA quantity, RNA-seq could be performed only on 14 specimens representing the two phenotypes Inflam^HIGH-Vasc^HIGH and Inflam^LOW Vasc^HIGH.

## Differential analysis of gene expression

Differentially expressed transcripts between RCM Inflam^HIGH and Inflam^LOW groups were determined using pairwise comparisons in edgeR[75–77] performing exact test (FDR < 0.05) on TMM-normalized read counts with filtering to remove lowly expressed transcripts (using filterByExpr argument in edgeR). R version 3.6.3 was used in analyses.

To determine sets of genes associated with RCM phenotypes, Co-Expression Modules identification Tool (CEMiTool)[28] was implemented in R to identify and analyze gene co-expression modules. Default parameters were used after assessing normal distribution of TMM normalized and log_2 transformed transcripts across samples (correlation method = Pearson, $R^2 > 0.8$, filtering $p$value = 0.1). Using CEMiTool[28], resulting gene modules were assessed for enrichment in RCM phenotype classes and inspected for expression patterns of individual genes within modules. Gene modules with significant enrichment in RCM phenotype were annotated for biological relevance using R interface with Enrichr[78–80] database (https://cran.r-project.org/web/packages/enrichR/vignettes/enrichR.html) for following curated gene sets: Gene Ontology (GO), pathway enrichment in curated MsigDB Hallmark gene set[81,82], cell/tissue-specific gene expression profiles using GTEx Tissue and Human Gene Atlas[83]. To discover module enrichment for disease processes, enrichment in GTEx Aging and OMIM repositories were performed as well. Gene-gene interactions to discover module gene connections with functional gene regulatory networks defined for T-lymphocytes, macrophage, skin, and blood were generated using gene pairs defined from TissueNexus[29] (https://www.diseaselinks.com/TissueNexus/index.php). Interaction data was used in CEMiTool[28] to identify overlap with module genes, discover network hub genes, and visualize network interactions. Network hub genes were used as input in TissueNexus database, selecting for appropriate tissue/cell type, to query

top-connected neighbors followed by GO enrichment analysis using GOTermFinder. Resulting top 5 enriched GO terms (adj $p$value< 0.05) were reported for M2 and M5. Eigen gene values summarizing expression of all genes within modules M2 and M5 were correlated (method = Spearman) with TiME phenotypes to interpret disease relevance and visualized using a correlation matrix along with scatterplot reporting significant $R$-value using Hmisc[84] package in R (https://cran.r-project.org/web/packages/Hmisc/Hmisc.pdf). To inspect major sources of variation in gene expression contributing to RCM phenotypes, principal component analysis (PCA) was performed using prcomp function and confidence ellipses in PCA were generated using factoextra[73] package in R (https://rpkgs.datanovia.com/factoextra/index.html).

## CIBERSORTx analysis

CIBERSORTx was used for the immune cell analysis to delineate immune subsets using 547 genes for 22 immune cell types[85]. Transcript per million values were used as input. CIBERSORTx chooses the record with the highest mean expression across the mixtures during analysis. The gene expression file with 14 cases was uploaded to CIBERSORTx as a mixture file, and CIBERSORTx was run with the following options: relative and absolute modes together, LM22 signature gene file, 100 permutations, and quantile normalization disabled. Sample distance matrix resulting from immune cell distribution and k-means clustering were used to interpret CIBERSORTx output.

## Immunohistochemistry

IHC for CD1a, CD68, CD3 and CD20 was performed on Bond Rx system (Leica Biosystems, US). The protocol for the Bond Rx platform included heat retrieval followed by primary antibody incubation (Santa Cruz Biotech, US). Polymer detection was through DAB Kit (Leica Biosystems, catalog #DS9800). For the dual CD3/CD20 IHC sequential staining, we performed heat retrieval, incubation with primary antibodies (anti-CD3, anti-CD20) followed by polymer detection kits (Leica Biosystems, US, catalog #DS9800, DS9390). The list of antibodies and their dilutions have been reported in Supplementary Methods as Table S1. The IHC slides were digitized on a slide scanner (Aperio Imagescope, Leica Biosystems, US).

## Multiplexed immunofluorescence and analysis

The Opal '7-color manual IHC kit' (Akoya Biosciences, Marlborough, MA) was used to conduct IF staining on the autostainer LabSat Research autostainer (Lunaphore Technologies, Tolochenaz, Switzerland). The stains were visualized using a Vectra Polaris Automated Quantitative Pathology Imaging System (Akoya Biosciences, Marlborough, MA). The quantitative multiplexed IF results were derived using the image analysis platform HALO version 2.3 (Indica Labs, Albuquerque, NM, USA). Depending on the size of the tissue, 1–12 (mean = 5.25) representative intra-tumoral and peritumoral region-of-interests (ROIs) were chosen, focusing on areas of immune infiltrates. Cell counts were summed up by compartment for every patient and values were normalized by DAPI + cells. Twenty-four specimens were available for evaluation (tissue quality was heavily deteriorated precluding analysis in 3 samples). Nuclear segmentation parameters were set using DAPI as the reference channel, then an analysis mask was overlaid on the image allowing the software to locate and segment each cell. Using a combination of the analysis mask and the scanned slide image, minimum intensity thresholds are set for each channel. Once the thresholds are set the analysis outputs a percentage of positive cells based on the number of cells positive for a specific phenotype and the number of DAPI + cells. All antibodies were diluted in Antibody diluent/block (ARD1001EA, Akoya Biosciences, Marlborough, MA, US). The list of antibodies and their dilution have been included in Supplementary File as Table S1.

## IHC evaluation quantification

Each immune marker was quantified using Positive Pixel counting algorithm (Aperio, Leica Biosystems, IL, US)[86]. Thresholding was performed on brown, pink, and total brown and pink areas, and total area was determined by hematoxylin-stained area. Integrated positive pixel area was used to compute the relative proportion of cells. Parameters for threshold, hue, and saturation were kept constant across all patient specimens.

## Immunophenotyping

Freshly excised 3 mm punch biopsies from 3 BCC lesions were collected in DMEM media. Tissue were transported to cell culture lab on ice and stored in DMEM media for 24–48 h at 4 °C. Cell suspensions were generated according to the following protocol[87]. Cells were processed for surface labeling with anti-CD3, anti-CD45, anti-CD4, and anti-CD8 antibodies. Live cells are distinguished from dead cells by using the fixable dye eFluor506 (eBioscience, Thermo Fisher Scientific, MA, USA). They were further permeabilized using a FOXP3 fixation and permeabilization kit (eBioscience, Thermo Fisher Scientific, MA, USA) and stained for Ki-67, FOXP3, and granzymeB. Data were acquired using the Aurora Five Laser flow cytometer (Cytek Biosciences, CA, US). Data were analyzed with FlowJo software version 10.5.3. (Tree Star Inc. OR, USA)[88].

## RCM feature quantification

**RCM data.** Individual images (0.75 × 0.75 mm) from stacks and single FOV frames from videos were used for automated quantification of immune cells, and vasculature, respectively. Machine learning-based immune cell quantification was explored on 1026 frames from 93 lesions (skin cancers and rashes) in 74 patients (34-females, 40-males; age range 30–88 years). Each case contributed 5-27 independent images. For vascular feature quantification, 438 single FOV videos (39, 813 frames). Each lesion contributed 1-31 videos.

**Machine learning for immune cells.** A pixel-wise segmentation model was trained for 4 important morphological patterns (dendritic cells, macrophages, leukocytic round-ellipsoid cells and miscellaneous immune cells). We binned them into 2 classes as Class 1: Dendritic cells and macrophages, Class 2: round-ellipsoid leukocyte-like cells. We labelled a third class called background comprising of areas that did not contain any of the immune cell patterns. A total of 1026 RCM images from 92 lesions were labelled pixelwise for these 3 classes in a non-exhaustive manner, where we only labelled examples of these patterns (Fig. S6a). A total of 9% of the pixels were labelled (6% Class 1, 3% Class 2), 91% was used as Class 3. We trained a 3 class UNet[89]segmentation model using the MONAI framework[90]. We used 926 images for training and 100 independent images for testing the model. Based on our former studies[59,91], we downsampled the RCM images to 256 by 256 pixels (corresponding to 2 μm resolution) for the sake of computational efficiency. We used a learning rate of 5e-2, batch size of 64, and SGD optimizer with Nesterov momentum. We also used image augmentation such as random rotation, flipping, elastic-affine deformation, intensity scaling, to increase the training dataset size. The model is trained for 90 epoch using DICE loss. After 90 epochs we did not see any improvement in the loss. We found a Dice similarity coefficient of 0.72 for these 3 classes (Table 2).

**Vascular features.** For vascular feature quantification, we used 438 single FOV videos (39, 813 frames) from 48 cases. For all video frames, we used a two-step image stabilization procedure to remove the significant motion found in each movie segment. Firstly, a linear pre-alignment is performed to minimize large-scale motion in FIJI[92] using the SIFT feature plugin Plugins->Registration-> Linear Stack Alignment with SIFT and default parameters. Stabilized images are then automatically cropped in MATLAB (mathworks.com) to remove

black background and include only areas within the FOV during the entirety of the movie segment. The crop rectangle is computed automatically by iteratively removing the row or column of pixels which contains the most blank pixels in a temporal min image until all outer edge rows and columns that contain more than three-quarters blank pixels are removed. We then performed a second custom nonlinear stabilization in MATLAB to remove large-scale tissue deformations over time. Frame t + 1 first has its histogram equalized to match frame t and then is aligned to frame t using the imregdemon procedure with four pyramid levels and steeply decreasing iterations of alignment at successively finer scales (iterations, [100,50,10,1]). Frame t + 2 is then aligned with the transformed frame t + 1 and so on. Cropping of all regions not in view throughout the movie was performed again via the same procedure.

**Blood vessel segmentation.** We performed manual segmentation of blood vessels using an open-source segmentation platform called 3D Splicer (https://www.slicer.org/)[93] on 25 randomly selected videos. Two videos were discarded from analysis due to extreme Z-motion. We processed the remaining 23 videos to display only every 10th frame to mimic the automated segmentation approach; each frame in the resulting file was manually segmented for blood vessels. We exported the entire video segmentation as a Nifti (.nii) file format and imported into MATLAB as a 3D image array, where consecutive images in the array correspond to consecutive frames in the RCM video. Ensuring that the consecutive frames are registered, our assumption for detecting the vessels was that the areas of high variation between consecutive frames correspond to vessels. In order to suppress the variation due to speckle noise in the RCM images, we first applied a gaussian smoothing filter (sigma = 1px). Then we applied a finite impulse response high pass filter (F = [0.5,−1,0.5]) and smooth out the extracted pixel-wise variation in time using a 7-by-7 median filter. We then subtracted the mean variation of each frame to eliminate the slowly varying areas, and obtain a variation map for the whole video by accumulating the variation over the entire video sequence. We finally applied otsu thresholding on the final variation map to locate the areas with vessels. To smooth the border of the vessels and clean out the noise in the segmentation, we applied morphological closing operation on the binary segmentation map and clean segmented areas smaller than 0.1% and larger 10% of the entire frame. Finally, we calculated Dice similarity coefficients by comparing manual and automated vessel segmentation (Fig. S6b, Table 2).

## Trafficking

**Background subtraction.** We estimated a background image for each frame as the median per pixel over a temporal window of 6 s centered on the current frame. Where movie temporal resolution differed, we adjusted the window in frames accordingly. This background estimate is subtracted out of the current frame, largely isolating moving cells on a dark background. We also tested alternative common methods for background subtraction, including sparse linear factorization methods as well as mean and min background estimates and dividing through by, rather than subtracting, background estimates, which desirably enhanced dim cells. None of these methods were found to provide satisfactory results on our data sets.

**Tracking.** We exported background subtracted images from MATLAB as 32-bit OME tiffs and imported into FIJI. Tracking is then performed in Trackmate[94] using DoG spot detection (subpixel = true; radius = 7.5pixels (7.5/1.33 = 5.63 μm); threshold = 1.6) and the LAP tracker with no splitting, merging or gap closing, and a max match distance of 20 pixels (20/1.33 = 15.03 μm). The tracklets found are then filtered in MATLAB to remove spurious tracklets corresponding to imperfectly removed background elements (this occurs particularly during changes in z during imaging) or tracks strung together from different fast

moving circulating blood cells while preserving the desired target population. Features used to measure tracklet desirability are detailed below. We set the thresholds quantitatively and automatically to maximize correspondence between automated results and manual counts on an initial training set of 40 movies (approx. 10% of overall data). We investigated three different temporal windows ranging from 3-5 frames (0.6 s, 0.8 s and 1 s) for total quantification of rolling, crawling and adherent cells. We adopted constrained optimization within a restricted range, although we also investigated fully independent threshold optimization (Fig. S6c). We found moderate-high correlation (0.79–0.82) during *first optimization* ((Table 2) following which trafficking was quantified on remaining videos. *Final validation* using manual counts on a subset of videos (~2.5% of total data) by two readers, we found high inter-reader concordance and high correlation (0.74–0.0.89) for different temporal windows (Fig. S6d, e). We selected temporal window 3 (0.6 s) for subsequent analysis to ensure inclusion of especially faster trafficking processes (rolling cells) in shorter blood vessels. The correlation was worse for videos with remnant motion after two-step motion minimization strategy, suggesting need for minimizing axial and lateral motion during data acquisition, and use of more efficient motion removal algorithms in future. The existing code can be found on Github and zenodo[95].

The Tracklet Parameters used are as follows:

Displacement = [15.41,16.92,16.92] μm or [20.5,22.5,22.5] pixels
Consistency = [58,58,58] degrees
Quality = [1.6,1.65,1.75] arbitrary units
Length = [0.6 s, 0.8 s or 1 s]

where,

Displacement – total displacement between tracklet start and end point, in pixels (tracklets with lower displacement are discarded)

Motion Consistency – average angle between the motion vector of the track at successive timepoints in degrees (tracklets with higher angular difference are discarded)

Quality – average quality of detections making up the tracklet as measured by Trackmate (lower average quality tracklets are discarded)

Length – duration in s of tracklet, in all cases this was set to the thresholds used in manual counts (shorter tracklets are discarded).

## Response to Immunotherapy analysis

We analyzed correlation of TiME features and phenotypes with response to topical TLRA imiquimod on 13 lesions on 8 patients (4 females, 4 males; age range 54–70 years). The patients were imaged at baseline (T0) and end of treatment. We performed HCPC clustering in R to identify clustering patterns based on response. To assign phenotypes, we projected these patients on the original BCC PCA model. Towards developing quantitative models for response prediction, we first performed linear regression modeling to quantitatively identify the predictor variables for response to imiquimod, and compared against the known "standard" which is tumor-infiltrating lymphocyte-like cells and intratumoral inflammation. In order to measure the predictive power of each feature, we trained predictive models in a leave-one-out cross-validation fashion and measured the model performance by inferring on the left-out test sample (out-of-bag estimates). This procedure was followed in an iterative manner, where we selected a single feature that gives the highest performance and added a new feature that provided the highest performance in each iteration. Model performance was measured by calculating specificity (the higher, the better) on the out-of-bag estimates and Akaike Information Criterion (the lower, the better) value of the model. In this way, the features were prioritized according to their predictive power. The best performance among the AIC prioritizing models was 85% sensitivity, 66% specificity and AIC = −15.06 with 8 variables while the best performance among the specificity prioritizing model was 71% sensitivity, 83% specificity and AIC = −15.06 with 13 variables. Moreover, we also examined the linear separability of (i) individual features by looking at

the histogram of feature values for each sample, and (ii) each pairwise feature combination by examining kernel density estimation plots.

## Statistical analysis

Data was collected using prospective patient imaging, no statistical method was used to predetermine sample size. Data from 1 patient was excluded due to motion blur during image acquisition during TiME phenotyping and all subsequent analyses. Being an observational, non-intervention study, the samples were not randomized. Further, the investigators were not blinded to allocation during experiments and outcome assessment. To elucidate agreements between two readers' manual evaluations for RCM features, Cohen's kappa coefficients were computed. The agreement regarding the extent of each feature presence between RCM and histology was quantified using linearly weighted Gwet's AC1. Unsupervised statistical clustering for phenotyping was performed using hierarchical clustering on principal components (HCPC) and principal component analysis (PCA) using PCs with largest eigenvalues that explained at least 95% of total variance. Mean, median and confidence intervals were computed for the multiplexed IHC/IF data analyzed and significance was estimated using either two-tailed unpaired Mann–Whitney test or Kruskal-Wallis test corrected for multiple group comparisons by Dunn's method. Differentially expressed transcripts between RCM Inflam[HIGH] and Inflam[LOW] groups were determined using pairwise comparisons in edgeR performing quasi-likelihood F-test (FDR < 0.05) on TMM-normalized read counts with filtering to remove lowly expressed transcripts (using filterByExpr argument in edgeR). Co-Expression Modules identification Tool (CEMiTool) was used to identify and analyze gene co-expression modules using default parameters after assessing normal distribution of TMM normalized and $\log_2$ transformed transcripts across samples (correlation method = Pearson, $R^2 > 0.8$, filtering $p$value = 0.1). Two-tailed Mann–Whitney tests were used to evaluate $p$-value for CIBERSORTx immune cell proportions, and prevalence of TiME features across responders and non-responders. Spearman correlation was used to estimate correlation between quantified RCM features and immune-related, trafficking-related and vascular-related genes. Linear regression modeling and leave-one-out estimates were used for developing prediction models of treatment response. Model performance was evaluated by specificity and Akaike information criterion.

## Reporting summary

Further information on research design is available in the Nature Research Reporting Summary linked to this article.

# Data availability

The RNA-expression datasets generated during this study have been made publicly available on Gene Expression Omnibus (GEO ID: GSE181037). Publicly available datasets used in this study include MsigDB [https://www.gsea-msigdb.org/gsea/msigdb/], gene ontology [http://geneontology.org/], GTex [https://gtexportal.org/home/] and OMIM repository [https://www.ncbi.nlm.nih.gov/omim]. De-identified raw images will be freely available upon request. Requests will be considered for 10 years after the publication of this article. As per MSKCC guidelines, a data sharing agreement and ethical permissions will have to be necessarily set up with requesting colleagues and their institutions. Source data are provided with this paper.

# Code availability

Custom image and data analysis scripts for FIJI and MATLAB developed for quantification of imaging data are available on GitHub [https://github.com/mskccmccf/TiME-analysis] or Zenodo [https://doi.org/10.5281/zenodo.6712717] and the RNA-seq analysis pipeline is available on GitHub [https://github.com/mskccmccf/TiME-analysis/tree/main/RNA_Seq_Analysis]. Other publicly available R packages used for the

gene expression and statistical analysis have been cited throughout the manuscript.

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

## Acknowledgements

We would like to acknowledge Dr. Anjali Rajadhyaksha for assistance with experimental planning and scientific discussion. We would also like to acknowledge Dr. Sanjee Abeytunge, Dr. Veronica Rotemberg and Dr. Mario Lacouture for critical feedback. We also acknowledge MSKCC Cores: Integrated Genomics Operation Core, funded by the NCI Cancer Center Support Grant (CCSG, P30 CA08748), Cycle for Survival, and the Marie-Josée and Henry R. Kravis Center for Molecular Oncology, Molecular Cytology Core Facility, Pathology Core, Flow Cytometry Core. In addition, Ms. Cassidy Cobbs, Ms. Marina Asher, Mr. Afsar Barlas, Mr. Eric Chan, Mr. Stephen Wilson and Mr. Reza Afzalneia for assistance with experimental planning, data analysis, tissue processing and immuno-histochemical staining. The authors would also like to acknowledge funding support from the following centers:

NIH/NCI Cancer Center Support Grant P30 CA008748 (MSKCC)
NIH/NIBIB R01EB028752 and NIH/NCI R01240771 (MR)
Melanoma Research Alliance (AS) (https://doi.org/10.48050/pc.gr.141697, https://doi.org/10.48050/pc.gr.89583)
The Chan-Zuckerberg Initiative (Anthony S)
Swiss National Science Foundation (LK)
Roux Institute and Harold Alfond Foundation (CB)
Ludwig Institute for Cancer Research, Sephora, Hazen Polsky Foundation, the Parker Institute for Cancer Immunotherapy and Swim Across America (Merghoub/Wolchok Lab).

## Author contributions

Conceptualization: A.S., K.K., M.C., M.G., C.A.F., S.G., A.R., A.H., L.D., M.P., A.M., C.J.C., T.M., M.R. Methodology development: A.S., K.K., L.K., A.H., C.B., T.T., A.S., P.C., S.Y., P.M., M.C., C.N.D., S.D., C.J.C. Imaging and histopathology: A.S., M.C., C.N.D., P.K., M.G., C.A.F., S.G., G.P., K.K., P.G., M.P., W.P. Molecular analyses: A.S., L.K., C.B., A.H., T.T., N.Y., S.L., W.P. Data analysis and visualization: A.S., K.K., L.K., C.B., A.H., T.T., A.S., A.A., M.L., K.K., P.K., A.W.W., G.P., P.C., S.Y., P.M., N.K., S.D., N.Y., S.L. Funding acquisition: L.D., T.M., M.R. Supervision: M.C., M.G., C.A.F., S.G., A.H., L.D., P.G., A.R., A.M., M.P., C.J.C., T.M., M.R. Writing: A.S., K.K., L.K., C.B., A.H., A.S., A.A., M.L., A.W.W., N.K., S.D., A.M., C.J.C., M.R.

## Competing interests

M.G. is a consulting investigator for DBV technologies; research consultant: Dermatology Service, MSKCC. Christi Alessi-Fox: employee of and owns equity in Caliber I.D., manufacturer of the VivaScope RCM. Dr. Rossi: Mavig (travel accommodation), Merz, DynaMed, Canfield Scientific, Evolus, Biofrontera, QuantiaMD, Lam Therapeutics, Cutera (consultant); Allergan (advisory board). A.H. : consultant to Canfield Scientific and an advisory board member of Scibase. L.D. : cofounder and holds equity in IMVAQ Therapeutics, patents on applications related to work on oncolytic viral therapy (US 20220056475 A1: recombinant poxviruses for cancer immunotherapy; US 20180236062 A1: use of inactivated nonreplicating modified vaccinia virus ankara (mva) as monoimmunotherapy or in combination with immune checkpoint blocking agents for solid tumors). A. M.: honorarium for dermoscopy lectures (3GEN), royalties for books/book chapters, dermoscopy equipment for testing, payment for organizing and lecturing (American Dermoscopy Meeting). C-S.J.C. : research funding from Apollo Medical Optics, Inc. Milind Rajadhyaksha: was employee of and owns equity in Caliber I.D. VivaScope is the commercial version of a laboratory prototype he developed at Massachusetts General Hospital, Harvard Medical School. T.M. has acted as a consultant for Immunogenesis, Immunos Therapeutics, Daiichi Sankyo, Leap therapeutics and Pfizer, has received research support from Adaptive Biotechnologies, Aprea, Bristol Myers Squibb, Infinity Pharmaceuticals, Kyn Therapeutics, Leap Therapeutics, Peregrine Pharmaceuticals and Surface Oncology, is a cofounder of and holds an equity in IMVAQ Therapeutics and is listed as a co-inventor on patents relating to the use of oncolytic viral therapy, alphavirus-based vaccines, antibodies targeting CD40, GITR, OX40, PD-1 and CTLA-4 and neo-antigen modelling (US 20220056475 A1: recombinant poxviruses for cancer immunotherapy; US 20210179714 A1: Inhibition of CTLA-4 and/or PD-1 For Regulation of T Cells; US 20200232040 A1: neoantigens and uses thereof for treating cancer; US 20200113984 A1: Alphavirus Replicon Particles Expressing TRP2; US 20180236062 A1: use of inactivated nonreplicating modified vaccinia virus ankara (mva) as mono-immunotherapy or in combination with immune checkpoint blocking agents for solid tumors). The remaining authors declare no competing interests.

## Additional information

Aditi Sahu [1] ✉, Kivanc Kose [1], Lukas Kraehenbuehl [2], Candice Byers [3,4], Aliya Holland[2], Teguru Tembo[1,5], Anthony Santella[1], Anabel Alfonso[1], Madison Li [1], Miguel Cordova[1], Melissa Gill[5,6,7], Christi Fox[8], Salvador Gonzalez[7], Piyush Kumar [9], Amber Weiching Wang[10], Nicholas Kurtansky [1], Pratik Chandrani [11], Shen Yin[1], Paras Mehta[1], Cristian Navarrete-Dechent[1,12], Gary Peterson[1], Kimeil King[1], Stephen Dusza [1], Ning Yang [1], Shuaitong Liu[1], William Phillips[1], Pascale Guitera[13,14], Anthony Rossi[1], Allan Halpern[1], Liang Deng [1,15], Melissa Pulitzer[1], Ashfaq Marghoob[1], Chih-Shan Jason Chen[1], Taha Merghoub [1,2,15] & Milind Rajadhyaksha [1] ✉

[1]Memorial Sloan Kettering Cancer Center, New York, NY, USA. [2]Parker Institute for Cancer Immunotherapy, Ludwig Collaborative and Swim Across America Laboratory, Human Oncology and Pathogenesis Program, Memorial Sloan Kettering Cancer Center, New York, NY, USA. [3]Roux Institute, Northeastern University, Portland, ME, USA. [4]Department of Electrical and Computer Engineering, Northeastern University, Boston, MA, USA. [5]SUNY Downstate Health Sciences University, Brooklyn, NY, USA. [6]Department of Clinical Pathology and Cancer Diagnostics, Karolinska University Hospital Solna, Stockholm, Sweden. [7]Faculty of Medicine and Health Sciences, University of Alcala, Madrid, Spain. [8]Caliber Imaging and Diagnostics, Rochester, NY, USA. [9]Icahn School of Medicine at Mount Sinai, New York, NY, USA. [10]ORIC Pharmaceuticals, San Francisco, USA. [11]Tata Memorial Hospital, Mumbai, India. [12]Pontificia Universidad Católica de Chile, Santiago, Chile. [13]Sydney Melanoma Diagnostic Center, Sydney, NSW, Australia. [14]Melanoma Institute Australia, Wollstonecraft, NSW, Australia. [15]Weill Cornell Medicine, New York, NY, USA. ✉e-mail: aditisahu@gmail.com; rajadhym@mskcc.org

