## [Peer Review File · Nature Communications]

This manuscript has been previously reviewed at another journal that is not operating a transparent peer review scheme. This document only contains reviewer comments and rebuttal letters for versions considered at *Nature Communications*.

Reviewers' comments:

Reviewer #1 (Remarks to the Author): Expert in imaging

Summary:

This paper has sought to address the issue of patient stratification to improve cancer response to immunotherapy using in vivo methods of tumour phenotyping with reflectance confocal microscopy (RCM). The authors obtained RCM images of the tumour-immune microenvironment (TiME) from different tumor lesions, classified tumour types based on the prevalence of vasculature and inflammation and correlated the morphological features in RCM images with RNA sequencing data, as well as assessed the subsequent response to immunotherapy.

Strengths of the paper:

The authors proposed a unique method for combining in vivo reflectance confocal microscopy (RCM) with tumor type classification for improved clinical treatment. An approach like this allows clinicians to make appropriate real-time judgments about patient treatment. The originality of the study, as well as the team's competence in reading RCM pictures to identify the essential characteristics of vascularisation and inflammation in the TiME, are the paper's main strengths, as demonstrated by the reasonably excellent agreement between 2 RCM readers and correlation with histology.

Major Points:

- The way the patients were segregated as described in text (lines 115-118) and Figure 1 is confusing as it appears as if the images were segregated by Machine-learning trained algorithms into vasculaturehi/lo and inflammationhi/lo categories. Was the PCA analysis in Figure 1B for the 27 patients carried out on manually annotated images, or were these images scored and classified by the trained Machine-Learning algorithm? From the manuscript, it appears that the machine-learning algorithm is only applied later for the existing Figure 3.
- Figures 3 and 4 are not in order, please swap them back. For the existing Figure 4B, it would be worthwhile discussing why there could be responders to immunotherapy with medium/high levels of vasculature.
- For existing Figure 3, what is the difference between "Trafficking-optimization" and "Trafficking - validation"? Vessel segmentation also appears to exhibit the highest variation with a Dice coefficient of 0.29 -0.78. Is there a way to improve this, and is this inherent in difficulty in interpreting dark regions within the image as vessels?
- The trained Machine learning algorithm will be important for helping clinicians to apply this novel approach within the clinic. Would this be made available to the medical community?
- Have the authors imaged normal skin as control?

Minor points:

- To add in Lines 156-157: "Higher leukocyte trafficking, stromal vessels, and stromal macrophages were present in 50%, 100% and 86% of the non-responders respectively"

Reviewer #2 (Remarks to the Author): Expert in vascular imaging

The paper by Sahu et al. proposes a novel and non invasive in vivo tumor immune microenvironment (TiME) phenotyping based on the use of reflectance confocal microscopy. They identify four major phenotypes and established correlation with inflammatory, angiogenic, trafficking and tumor-intrinsic signature. A further aim is to determine the relative importance of the phenotypes in predicting response to topical immunotherapy. The model used is basal cell carcinoma.

The subject is very interesting mainly because of the combination of dynamic vascular and immune features, which may favor better patient stratification for immunotherapy. Why did the authors use BCC and not melanoma as a model? The reason should be explained. Melanoma presents a more pressing challenge in defining the immunoscore. The proposal to use BCC (which is easier to treat) as a model for response to therapy may not provide a crucial and solid basis for patient stratification based on TiME phenotypes in a more aggressive skin tumor. Could the authors provide some preliminary information on TiME characteristics obtained by RCM for a small cohort of melanoma patients?

I have several concerns as reported by the following summary.

From a general point of view, the weakest part of the manuscript is the Results section. The introduction and part of the discussion are well written, the Methods section is very detailed, but

the results are described as general findings without highlighted evidence (e.g., first paragraph, lines 124-126: what coming out from the correlation with the total area of CD3 positive T cells and the area of TLS structures? Fig. 1C is not sufficient to provide a clear definition between the four phenotypes based on the total area of T cells and the area of TLS. Moreover, Figure S3 is not self-explanatory either). By the way, Figure S3 shows representative staining for both BCC and SCC (although this is not stated). Why actually? The paper focuses on BCC and no further information or comparison is provided for SCC. Fig. S8 provides some sort of validation for a likely correlation between RCM phenotypes and immune activation. This is one result, but only in the Discussion is the figure mentioned. Indeed, more detailed results are included in the Discussion, but are completely absent from the relevant section. This "selection" makes it difficult to read and does not make the results clear at all.

There are many errors in the numbering of the Figures and associated data. Fig. S4E does not exist (line 142), nor does Fig. S4D (line 146), Fig. 4SF (line 147). Line 140, Fig. S5C and not Fig S4C. What are the different cohorts in Fig. S5 (line 150)? Fig. 3A is actually Fig. 4A; in any case, where can I find a description of Figure 3 indicating quantification of TiME features and correlation with gene expression with a corresponding comment? Maybe in the figure legend? Fig S6C is missing. Line 173: the corresponding figure is not Fig 4B as indicated in the text (I suspect Fig. 3B). All these examples indicate very low accuracy in the presentation of the data and suggest that there might also be misunderstanding and misleading in clustering, interpretation and analyses of the data. For example, why is the number of points in the PCA analyses greater than the number reported in the corresponding heatmap (BCC n= 13) (Fig. S4A)? Fig. S4C, heatmap: BCCs are numbered differently than in A and B. Do they belong to different patients? Why were the different GEO datasets not performed on the same group of patients (13 in A, 5 in B, 12 in C)? Some Figures are not well resolved, so they are not legible (fig S3, Fig S4, Fig S7, Fig 2C); some others are not accurately drawn: Fig.2, letter B is missing from the graph; Fig. S3A-D, purple and brown label..., the scale bar is missing.

In summary, the lack of clarity and the reader's effort to find the results throughout the manuscript detract from the significance of the authors' findings.

Clustering on PCA data revealed four major phenotypes. Detailed clinicopathological data are lacking. Is there a correlation with staging or other features?

Minor points

Table 1: How did the pathologists rate "trafficking" for the match?

Fig. 2E: the color bar on the left should be defined. Fig. 2, Legend: are C and D truly representative? Are 3 vessels vs 2 vessels a significant difference?

Table S1: How were index and scattering coefficient calculated? References are referred to...?

Reviewer #3 (Remarks to the Author): Expert in BCC imaging

The paper titled "Cellular-level phenotyping of tumor-immune microenvironment (TiME) in patients in vivo reveals distinct inflammation and endothelial anergy signatures" is a quite interesting manuscript. The topic of this manuscript falls within the scope of Nature Communication.

The Authors presented a novel tumor immune microenvironment (TiME) phenotyping in vivo in patients with non-invasive spatially-resolved cellular-level imaging based on endogenous contrast. They determined four major phenotypes with variable prevalence of vasculature (Vasc) and inflammation (Inf) features: VaschiInfhi, VaschiInflo, VascloInfhi and Vascmed/hiInflo. The Authors showed that the VaschiInfhi phenotype correlates with high immune activation, exhaustion, and vascular signatures while VaschiInflo with endothelial anergy and immune exclusion.

The data has been provided with vigorous statistical analysis. The Authors have presented sufficient data. The appropriate tables and figures have been provided. The article is easy to read and logically structured. The Authors used appropriate statistic methods. The Authors have presented sufficient data. The appropriate tables and figures have been provided. The methods are adequately described. The conclusions are consistent with presented evidence and arguments.

References are up to date and complete.

In my opinion the paper may be published in presented form.

Reviewer #4 (Remarks to the Author): Expert in tumour microenvironment and gene expression

General comments

The manuscript by Sahu and colleagues describes an in vivo phenotyping method for investigating skin cancer. The authors used reflectance confocal microscopy imaging to characterize skin tissue from 27 patients of which 13 received topical immunotherapy. Four phenotypes were determined based on vasculature and inflammation features and correlation analysis was carried out using bulk RNA sequencing data from a subset of the patients. Response to topical immunotherapy was associated with one specific phenotype.

There are several concerns with the manuscript. First, and most important, given the small number of patients, the lack of thorough validation, and the lack of benchmarking with other techniques, the current study is too preliminary and needs to be elaborated further. Second, there are a number of methodological and technical issues that need to be addressed (see below). And third, the manuscript requires extensive rewriting in order to: 1) provide more and important details, and 2) reframe some of the statements.

Specific comments

1. The stratification of the patients in four groups seems arbitrary. In the PCA analysis (Figure 1B) there is no clear separation of the patients in distinct groups. Moreover, the Vascmed/hi/Inflo group has only two patients! It is also not clear why was Vascmed/hi/Inflo selected and not Vasclo/Inflo.
2. In-depth validation using extensive histopathology data (IHC and/or FACS) to assign the specific cell types to the suggested features, and hence patient groups is of utmost importance but is missing.
3. How are other approaches like dynamic optical coherence tomography or optical frequency domain imaging performing, etc. compared to the presented approach? The authors described these approaches nicely in the discussion, but there is no benchmarking data with at least some of these.
4. The choice of the CIBERSORT algorithm is questionable. The original CIBERSORT algorithm was developed based in microarray data which has different dynamic range. The authors should use the improved version CIBERSORTx (Newman et al., Nat Biotech 2019) or one of the alternative deconvolution methods (see review PMID: 31515541).
5. The analysis of the response data is not convincing. Only 4 out of 7 responders belonged to the Vasclo/Infhi group. Clearly, with this small numbers a ROC curve cannot be made.
6. The mode of action of immune checkpoint blockers is totally different from that of Imiquimod, and hence, the results obtained in this study cannot be extrapolated to therapy with immune checkpoint blockers. Hence, major part of the manuscript has to be rewritten and/or deleted (e.g. second paragraph on page 7).

Response to reviewers' comments:

Reviewer #1 (Remarks to the Author): Expert in imaging

Summary:

This paper has sought to address the issue of patient stratification to improve cancer response to immunotherapy using in vivo methods of tumour phenotyping with reflectance confocal microscopy (RCM). The authors obtained RCM images of the tumour-immune microenvironment (TiME) from different tumor lesions, classified tumour types based on the prevalence of vasculature and inflammation and correlated the morphological features in RCM images with RNA sequencing data, as well as assessed the subsequent response to immunotherapy.

Strengths of the paper:

The authors proposed a unique method for combining in vivo reflectance confocal microscopy (RCM) with tumor type classification for improved clinical treatment. An approach like this allows clinicians to make appropriate real-time judgments about patient treatment. The originality of the study, as well as the team's competence in reading RCM pictures to identify the essential characteristics of vascularisation and inflammation in the TiME, are the paper's main strengths, as demonstrated by the reasonably excellent agreement between 2 RCM readers and correlation with histology.

We thank the reviewer for their encouraging comments.

Major Points:

- The way the patients were segregated as described in text (lines 115-118) and Figure 1 is confusing as it appears as if the images were segregated by Machine-learning trained algorithms into vasculature_{hi/lo} and inflammation_{hi/lo} categories. Was the PCA analysis in Figure 1B for the 27 patients carried out on manually annotated images, or were these images scored and classified by the trained Machine-Learning algorithm? From the manuscript, it appears that the machine-learning algorithm is only applied later for the existing Figure 3.

We will clear the confusion. The PCA in Figure 1 (Figure 2 in the revised manuscript) was carried out on manually annotated images, since the spatial context of TiME features, an important component for the classification, was present only in manual evaluation. The machine-learning algorithms in this study have been trained only in feature recognition. We have clarified this in the Methods section. The changes in the revised manuscript can be found in the following section:

Section: Methods, Page number: 18, Line number: 442

- Figures 3 and 4 are not in order, please swap them back. For the existing Figure 4B, it would be worthwhile discussing why there could be responders to immunotherapy with medium/high levels of vasculature.

We have corrected the Figure numbering and we will avoid repeating such mistakes. An explanation regarding the response to immunotherapy with respect to vasculature has been included in the revised manuscript.

Section: Discussion, Page number: 13, Line number: 306

- For existing Figure 3, what is the difference between "Trafficking-optimization" and "Trafficking-validation"? Vessel segmentation also appears to exhibit the highest variation with a Dice coefficient of 0.29 -0.78. Is there a way to improve this, and is this inherent in difficulty in interpreting dark regions within the image as vessels?

For trafficking, we first selected about 10% of the videos (n=40) which were manually counted for instances of trafficking. The parameters for trafficking quantification were selected using these values as the ground truth and run on TrackMate, FIJI. The accuracy of classification based on these parameters for the 40 videos is listed under “trafficking-optimization”. This set of parameters was tested on the remaining 90% of the data and validated by two readers on a small subset of videos, presented as “trafficking-validation”.

We completely agree with the reviewer about the Dice coefficient for vessel segmentation demonstrating large variations. We recognized that some anatomical locations e.g. nose and accidental motion in the Z axis (axial motion) during video acquisition by the hand-held probe resulted in the most challenging data for vessel segmentation. Mitigating these by employing tissue coupling during video acquisition and advanced frame stabilization tools are part of our ongoing efforts.

- The trained Machine learning algorithm will be important for helping clinicians to apply this novel approach within the clinic. Would this be made available to the medical community? - Have the authors imaged normal skin as control?

We completely agree with the reviewer regarding the machine learning tools and their importance for clinical translation of our approach. This, in fact, was the motivation behind the efforts in optimizing automated quantification. We have made all our codes available to the scientific and medical community through GitHub (link in the manuscript).

Normal skin (outside lesion) is routinely imaged as control. However, since normal skin lack TIME features, it has been excluded from the analysis. We have presented some representative normal images in the supplementary data.

Section: *Supplementary Figures, Figure number:* Figure S1d.

Minor points:

- To add in Lines 156-157: “Higher leukocyte trafficking, stromal vessels, and stromal macrophages were present in 50%, 100% and 86% of the non-responders respectively”

We have made this change in the revised manuscript.

Section: *Results, Page number:* 9, **Line number:** 214

Reviewer #2 (Remarks to the Author): Expert in vascular imaging

The paper by Sahu et al. proposes a novel and noninvasive in vivo tumor immune microenvironment (TiME) phenotyping based on the use of reflectance confocal microscopy. They identify four major phenotypes and established correlation with inflammatory, angiogenic, trafficking and tumor-intrinsic signature. A further aim is to determine the relative importance of the phenotypes in predicting response to topical immunotherapy. The model used is basal cell carcinoma.

The subject is very interesting mainly because of the combination of dynamic vascular and immune features, which may favor better patient stratification for immunotherapy. Why did the authors use BCC and not melanoma as a model? The reason should be explained. Melanoma presents a more pressing challenge in defining the immunoscore. The proposal to use BCC (which is easier to treat) as a model for response to therapy may not provide a crucial and solid basis for patient stratification based on TiME phenotypes in a more aggressive skin tumor. Could the authors provide some preliminary information on TiME characteristics obtained by RCM for a small cohort of melanoma patients?

We completely agree about the true need for this TiME phenotyping in aggressive cancers such as melanoma. In fact, after optimizing TiME features and biological correlation in BCCs, we have initiated studies on melanoma. As per the reviewer's suggestion, we have included preliminary TiME phenotyping on 13 melanoma lesions, validated with histopathology and CD3/CD20 IHC in the revised manuscript (Figure 5).

In the initial study, we used basal cell carcinoma (BCCs) as a preliminary model because of their i) higher incidence and ii) low aggressive nature. Thus, this helped us recruit sufficient numbers, while minimizing the chances of compromising patient care (The protocol for patient care is a bit more forgiving for BCCs than that for melanoma and other more serious skin conditions, which makes BCCs an excellent and implementable starting point for research on human patients). This proof-of-concept study in BCCs now sets the stage for verifying TiME phenotyping in melanoma, cutaneous lymphoma and oral squamous cell carcinoma which are routinely treated with immune checkpoint blockade therapies.

The changes in the revised manuscript can be found in the following section:

Section: Results, Page number: 8, Line number: 180

Section: Figures, Figure number: 5

I have several concerns as reported by the following summary.

From a general point of view, the weakest part of the manuscript is the Results section. The introduction and part of the discussion are well written, the Methods section is very detailed, but the results are described as general findings without highlighted evidence (e.g., first paragraph, lines 124-126: what coming out from the correlation with the total area of CD3 positive T cells and the area of TLS structures? Fig. 1C is not sufficient to provide a clear definition between the four phenotypes based on the total area of T cells and the area of TLS. Moreover, Figure S3 is not self-explanatory either). By the way, Figure S3 shows representative staining for both BCC and SCC (although this is not stated). Why actually? The paper focuses on BCC and no further information or comparison is provided for SCC. Fig. S8 provides some sort of validation for a likely correlation between RCM phenotypes and immune activation. This is one result, but only in the Discussion is the figure mentioned. Indeed, more detailed results are included in the Discussion, but are completely absent from the relevant section. This "selection" makes it difficult to read and does not make the results clear at all.

We have over-hauled the results sections to address the reviewer's concerns. In addition, we removed the SCC results which are not pertinent to this manuscript. Importantly, we have revised the Figure 1C (Figure 4b in the revised manuscript) to serve as a validation for RCM findings. We have also added multiplexed IF results for additional validation (Figure 4a).

Section: Results, Page numbers: 5-9

Section: Figures, Figure number: 2-7

There are many errors in the numbering of the Figures and associated data. Fig. S4E does not exist (line 142), nor does Fig. S4D (line 146), Fig. 4SF (line 147). Line 140, Fig. S5C and not Fig S4C. What are the different cohorts in Fig. S5 (line 150)? Fig. 3A is actually Fig. 4A; in any case, where can I find a description of Figure 3 indicating quantification of TiME features and correlation with gene expression with a corresponding comment? Maybe in the figure legend? Fig S6C is missing. Line 173: the corresponding figure is not Fig 4B as indicated in the text (I suspect Fig. 3B). All these examples indicate very low accuracy in the presentation of the data and suggest that there might also be misunderstanding and misleading in clustering, interpretation and analyses of the data. For example, why is the number of points in the PCA analyses greater than the number reported in the corresponding heatmap (BCC n= 13) (Fig. S4A)? Fig. S4C, heatmap: BCCs are numbered differently than in A and B. Do they belong to different patients? Why were the different GEO datasets not performed on the same group of patients (13 in A, 5 in B, 12 in C)? Some Figures are not well resolved, so they are not legible (fig S3, Fig S4, Fig S7, Fig 2C); some others are not accurately drawn: Fig.2, letter B is missing from the graph; Fig. S3A-D, purple and brown label...., the scale bar is missing. In summary, the lack of clarity and the reader's effort to find the results throughout the manuscript detract from the significance of the authors' findings.

We have corrected all the errors in Figure numbering, in main and supplementary figures.

While we agree that these inaccuracies can be concerning, we assure the reviewer that the main results have been thoroughly verified. Most of the analyses have been reproduced by multiple authors.

Clustering on PCA data revealed four major phenotypes. Detailed clinicopathological data are lacking. Is there a correlation with staging or other features?

We did not find any correlation of phenotypes with clinic-pathologic features such as BCC subtypes (superficial, nodular, infiltrative) (Figure 2a) and melanoma subtypes (melanoma in situ, superficial spreading melanoma) (Figure 5a). Similarly, other patient/tumor characteristics such as sun-exposure, age, gender and tumor ulceration also did not show any correlation with phenotypes. These findings have been included in the revised manuscript.

Section: Results, Page number: 5, 8 Line number: 109, 183

Section: Figures, Figure number: 2a, 5a

Minor points

Table 1: How did the pathologists rate “trafficking” for the match?

*Trafficking in H&E slides was identified as leukocytes attached to the endothelial wall called “stuck leukocytes” or occasionally leukocytes squeezing out of the vasculature undergoing diapedesis. The relative presence of such trafficking leukocytes in vessels over total vessels were graded by the pathologist. Examples of TiME features on histopathology, including leukocyte trafficking, have been included in **Figure S1** in the revised manuscript.*

Section: Supplementary Figures, Figure number: S1c

Fig. 2E: the color bar on the left should be defined. Fig. 2, Legend: are C and D truly representative? Are 3 vessels vs 2 vessels a significant difference?

We have reorganized the data on gene expression.

Section: Results, Page number: 5-6, Line number: 120

Section: Figures, Figure number: 3

Table S1: How were index and scattering coefficient calculated? References are referred to...?

We have decided to remove this table from the revised manuscript.

Reviewer #3 (Remarks to the Author): Expert in BCC imaging

The paper titled “Cellular-level phenotyping of tumor-immune microenvironment (TiME) in patients in vivo reveals distinct inflammation and endothelial anergy signatures” is a quite interesting manuscript. The topic of this manuscript falls within the scope of Nature Communication.

The Authors presented a novel tumor immune microenvironment (TiME) phenotyping in vivo in patients with non-invasive spatially-resolved cellular-level imaging based on endogenous contrast. They determined four major phenotypes with variable prevalence of vasculature (Vasc) and inflammation (Inf) features: VaschiInfhi, VaschiInflo, VasclInfhi and Vascmed/hilnflo. The Authors showed that the VaschiInfhi phenotype correlates with high immune activation, exhaustion, and vascular signatures while VaschiInflo with endothelial anergy and immune exclusion.

The data has been provided with vigorous statistical analysis. The Authors have presented sufficient data. The appropriate tables and figures have been provided. The article is easy to read and logically structured. The Authors used appropriate statistic methods. The Authors have presented sufficient data. The appropriate tables and figures have been provided. The methods are adequately described. The conclusions are consistent with presented evidence and arguments. References are up to date and complete. In my opinion the paper may be published in presented form.

We thank the reviewer for the encouraging feedback.

Reviewer #4 (Remarks to the Author): Expert in tumour microenvironment and gene expression

General comments

The manuscript by Sahu and colleagues describes an in vivo phenotyping method for investigating skin cancer. The authors used reflectance confocal microscopy imaging to characterize skin tissue from 27 patients of which 13 received topical immunotherapy. Four phenotypes were determined based on vasculature and inflammation features and correlation analysis was carried out using bulk RNA sequencing data from a subset of the patients. Response to topical immunotherapy was associated with one specific phenotype.

There are several concerns with the manuscript. First, and most important, given the small number of patients, the lack of thorough validation, and the lack of benchmarking with other techniques, the current study is too preliminary and needs to be elaborated further. Second, there are a number of methodological and technical issues that need to be addressed (see below). And third, the manuscript requires extensive rewriting in order to: 1) provide more and important details, and 2) reframe some of the statements.

We thank the reviewer for their critique. We agree with the reviewer about the need for validation and the manuscript re-writing. We have incorporated these changes in the revised manuscript (detailed below). We would also like to highlight the distinct patient cohort for each analysis: BCC phenotyping (n=27), melanoma phenotyping (n=13), topical immunotherapy (n=13).

Specific comments

1. The stratification of the patients in four groups seems arbitrary. In the PCA analysis (Figure 1B) there is no clear separation of the patients in distinct groups. Moreover, the Vascdmed/hi/Inflo group has only two patients! It is also not clear why was Vascdmed/hi/Inflo selected and not Vascdlo/Inflo.

The classification in Figure 1B (Figure 2f in the revised manuscript) is driven by the principal component vectors (shown in Figure 2b-e). We validated the patient stratification with another unbiased approach called hierarchical cluster analysis (HCA) which can be performed on principal components, called hierarchical clustering on principal components (HCPC) (Figure 2a). We agree with the reviewer that a separate category for only 2 patients is unnecessary and have clustered them as deemed fit by the HCPC unsupervised clustering approach. The changes in revised manuscript can be found here:

Section: Results, Page number: 5, Line number: 99

Section: Figures, Figure number: 2

2. In-depth validation using extensive histopathology data (IHC and/or FACS) to assign the specific cell types to the suggested features, and hence patient groups is of utmost importance but is missing.

We agree with the reviewer about the need for additional bench-marking of our novel TIME phenotypes. To address the reviewer's concern, we performed multiplexed IF experiments for T-cell, macrophage and immune checkpoint marker expression (PD1, PDL1, CD8, FOXP3, CD68) on BCC patients, based on our results from the CIBERSORTx analysis that indicated variability in the T-cell and macrophage populations.

Regarding validation on FACS, routine use of fresh tissue for research is precluded in the current skin cancer model because the primary lesions seen at our tertiary care cancer center tend to be small,

averaging 5-6 mm. Performing FACS, for research studies, on these smaller tissues is thus constrained by the requirements of standard of care, especially histopathological diagnosis. Nonetheless, we were successful in acquiring tissue from 3 patients, to support this feasibility study.

For phenotype validation, we relied primarily on multiplexed IF analysis, gene expression and dual CD3/CD20 IHC (to identify tertiary lymphoid structures known to be positively prognostic in several cancers, including melanoma).

Section: Results, Page number: 7, Line number: 156

Section: Figures, Figure number: 4

3. How are other approaches like dynamic optical coherence tomography or optical frequency domain imaging performing, etc. compared to the presented approach? The authors described these approaches nicely in the discussion, but there is no benchmarking data with at least some of these.

Dynamic OCT and OFDI are approaches optimized for imaging vasculature to about 1-2 mm depth in tissue. Their imaging resolution of 10-20 micron is insufficient for visualizing individual cells within the tumor microenvironment. In contrast, reflectance confocal microscopy affords nuclear-level and cellular-level resolution for imaging both cells and vasculature. In future, combining RCM imaging for cellular-level imaging of tumor, the immune microenvironment, and the vasculature along with dynamic OCT for deeper vessel imaging would help advance the approach. We have not performed any dynamic OCT/OFDI imaging in the current study, also because of non-availability of such a device at the center.

4. The choice of the CIBERSORT algorithm is questionable. The original CIBERSORT algorithm was developed based in microarray data which has different dynamic range. The authors should use the improved version CIBERSORTx (Newman et al., Nat Biotech 2019) or one of the alternative deconvolution methods (see review PMID: 31515541).

We regret the typographical error. The current analysis was indeed performed by CIBERSORTx.

5. The analysis of the response data is not convincing. Only 4 out of 7 responders belonged to the Vasclo/Infhi group. Clearly, with this small numbers a ROC curve cannot be made.

We agree with the reviewer that the sample size of the response data is very small to make any conclusive claims regarding phenotypic correlation with response. Thus, we present a very preliminary association between RCM TiME features, phenotypes and response to treatment. Modeling of TiME features using Akaike information criterion and specificity as the criteria revealed importance of both immune and vascular features in predicting response to treatment. This study sets the foundation for a larger study which will enable more robust ROC analysis for more representative sensitivity and specificity estimates.

6. The mode of action of immune checkpoint blockers is totally different from that of Imiquimod, and hence, the results obtained in this study cannot be extrapolated to therapy with immune checkpoint blockers. Hence, major part of the manuscript has to be rewritten and/or deleted (e.g. second paragraph on page 7).

We agree with the reviewer that the mode of response of imiquimod, which is a TLR-agonist, is very different from the immune checkpoint therapies targeting CTLA-4 and PD-1/PD-L1. Our study on BCCs and imiquimod as a model sets the stage for replicating in other more aggressive cancers such as

melanoma, cutaneous lymphoma and head/neck cancers that receive immune checkpoint therapies. We have made the necessary changes in the discussion as per the reviewer's suggestion.

Section: Discussion, **Page number:** 13, **Line number:** 306

REVIEWERS' COMMENTS

Reviewer #2 (Remarks to the Author):

The manuscript has been greatly improved with new data and figures. The Results section is now well detailed and written, thus easier to read.

I have only some concern about the quality of some figures and in particular of lack of accuracy as follows:

- Figure 4. a) the labels on the top of the images do not correspond to the "colored" markers map reported on the bottom. As an example, the column on the left shows also a magenta signal in addition to the orange one. The other column should indicate positivity for PD1 but the signal is light blue (CD68 according to coloured map). b) Even if easily comprehensible, no indication for purple and brown signal from immunohistology is provided.

- Figure 5. Arrows (yellow, light blue and red) should be indicated in the legend. In a) histology of the two phenotype are inverted. As in Figure 4b, no indication for purple and brown signal from immunohistology is provided.

Reviewer #4 (Remarks to the Author):

The authors addressed all the concerns in the original review satisfactorily. Most importantly, they carried out additional validation experiments using multiplexed IF and dual CD3/CD20 IHC assays. Furthermore, the methodological and technical issues were clarified and the manuscript was re-written.

REVIEWERS' COMMENTS

Reviewer #2 (Remarks to the Author):

The manuscript has been greatly improved with new data and figures. The Results section is now well detailed and written, thus easier to read.

We thank the reviewer for the encouraging remarks.

I have only some concerns about the quality of some figures and in particular of lack of accuracy as follows:

- Figure 4. a) the labels on the top of the images do not correspond to the “colored” markers map reported on the bottom. As an example, the column on the left shows also a magenta signal in addition to the orange one. The other column should indicate positivity for PD1, but the signal is light blue (CD68 according to coloured map). b) Even if easily comprehensible, no indication for purple and brown signal from immunohistology is provided.

We thank the reviewer for identifying this error: PD1 and CD68 had been inversed in the color bar. This has been corrected in the revised figure. Furthermore, the magenta appearance in left column CD8 panels is due to overlapping of DAPI and CD8, particularly in area of higher background staining for DAPI.

We have also specified the color code for the dual immunohistochemistry in the figure legend. The brown cells are CD3 T-cells while the purple/pink cells are CD20 B-cells.

- Figure 5. Arrows (yellow, light blue and red) should be indicated in the legend. In a) histology of the two phenotype are inverted. As in Figure 4b, no indication for purple and brown signal from immunohistology is provided.

We thank the reviewer for pointing this out: we have specified the arrow colors in the figure legend, corrected the order of the H&E images, and also specified the color code for the dual histochemistry in figure legend.

Reviewer #4 (Remarks to the Author):

The authors addressed all the concerns in the original review satisfactorily. Most importantly, they carried out additional validation experiments using multiplexed IF and dual CD3/CD20 IHC assays. Furthermore, the methodological and technical issues were clarified and the manuscript was re-written.

We thank the reviewer for their useful feedback and their positive remarks.